# Parental vaccination to reduce measles immunity gaps in Italy

Valentina Marziano[1]*, Piero Poletti[1], Filippo Trentini[1], Alessia Melegaro[2,3], Marco Ajelli[1,4], Stefano Merler[1]

[1]Center for Information Technology, Fondazione Bruno Kessler, Trento, Italy; [2]Department of Social and Political Sciences, Bocconi University, Milano, Italy; [3]Carlo F Dondena Centre for Research on Social Dynamics and Public Policy, Bocconi University, Milano, Italy; [4]Laboratory for the Modeling of Biological and Socio-Technical Systems, Northeastern University, Boston, United States

**Abstract** High-income countries are experiencing measles reemergence as the result of suboptimal vaccine uptake and marked immunity gaps among adults. In 2017, the Italian Government introduced mandatory vaccination at school entry for ten infectious diseases, including measles. However, sustainable and effective vaccination strategies targeting adults are still lacking. We use a data-driven model of household demography to estimate the potential impact on future measles epidemiology of a novel immunization strategy, to be implemented on top of the 2017 regulation, which consists of offering measles vaccine to the parents of children who get vaccinated. Model simulations suggest that the current vaccination efforts in Italy would not be sufficient to interrupt measles transmission before 2045 because of the frequency of susceptible individuals between 17 and 44 years of age. The integration of the current policy with parental vaccination has the potential to reduce susceptible adults by 17–35%, increasing the chance of measles elimination before 2045 up to 78.9–96.5%.
DOI: https://doi.org/10.7554/eLife.44942.001

*For correspondence:
marziano@fbk.eu

**Competing interests:** The authors declare that no competing interests exist.

## Introduction

The Global Measles and Rubella Strategic Plan 2012–2020 set the ambitious goal of eliminating measles in at least five World Health Organization (WHO) regions by 2020. Two years before the deadline, only the Americas have achieved measles elimination. Measles is endemic in 14 countries of the WHO European Region, including high-income countries such as Germany, Belgium, France, and Italy (*World Health Organization Regional Office for Europe, 2016*), and it still represents a major concern for public health.

In 2017, Italy experienced one of the largest measles outbreaks of the past decade in the European Region with four deaths and 5098 cases, 4042 of which were confirmed by positive laboratory results (*Italian National Institute of Health, 2017*; *European Centre for Disease Prevention and Control, 2018*). The highest incidence was observed in infants under one year of age. About 70% of the reported cases were older than 20 years, with a median age of 27 years (*Italian National Institute of Health, 2017*; *Filia et al., 2017*), suggesting that measles circulation in Italy is at least partially supported by transmission between adults. Significant immunity gaps in these age segments of the population have been highlighted by a serological screening of the population (*Rota et al., 2008*) and by recent modeling studies analyzing long-term processes that affect measles transmission dynamics in the Italian population (*Merler and Ajelli, 2014*; *Trentini et al., 2017*). The high fraction of measles-susceptible individuals of between 15 and 45 years of age is the result of past suboptimal routine vaccination coverage and the absence of major nationwide epidemics in recent decades, which allowed adolescents to escape both vaccination and natural infection (*Filia et al.,*

**eLife digest** Measles is one of the world's most contagious diseases causing thousands of deaths every year, despite a safe and effective vaccine being available since the 1960s. High rates of vaccination – about 95% of each age group – are required to eliminate measles, but national and global health agencies struggle to achieve high vaccination rates because some parents were and still are hesitant to vaccinate their children. As a result, large measles epidemics continue to occur even in countries with well-established vaccination programs.

In Italy, low vaccination rates year after year have resulted in large numbers of unprotected youth and adults. The country has recently introduced mandatory measles vaccination at school entry to improve vaccination coverage among children. Yet a high proportion of measles cases in Italy continue to occur in people over 20 years old, a situation that could be improved by immunization programs targeting adults. One approach would be to take advantage of the compulsory vaccination of children by offering parents the vaccine at the same time.

Marziano et al. used computer modeling to estimate how various vaccination scenarios would affect measles spread in Italy. Their models showed that current vaccination policies targeting school age children would be unlikely to eliminate measles before 2045. On the other hand, if 50% of parents were also vaccinated, elimination could be achieved by 2042, and as early as 2031 if 99% of parents agreed to vaccination.

Marziano et al. show that a parental vaccination campaign could reduce the population of adults susceptible to measles in Italy and help the country stop the spread of the disease. However, more research is needed to assess how feasible and sustainable this policy would be. Additional policies to increase vaccination against measles in adults could also help, but parental vaccination has a key advantage: it does not require active targeting to recruit parents, since they are already immunizing their children.

DOI: https://doi.org/10.7554/eLife.44942.002

*2017*; *Trentini et al., 2017*). In Italy, the first measles national immunization program was setup in 1983 with a single dose of measles vaccine being administered at 9 months of age. A second dose program was introduced in 1999. However, routine vaccination coverage remained below 80% until 2003, the year of approval of the Italian National Plan for the elimination of Measles and Congenital Rubella. Thereafter, vaccine uptake levels have progressively increased, even though a decrease in coverage has been detected in most recent years, possibly associated with vaccine hesitancy (*Filia et al., 2017*; *Merler and Ajelli, 2014*; *Giambi et al., 2018*). As a matter of fact, the national coverage reached a peak of 91% in 2010, which is well below the 95% threshold generally considered to be necessary for measles elimination (*Anderson and May, 1991*).

In July 2017, the Italian Government approved a regulation (119/2017) requiring parents to vaccinate their children before school entry against ten infectious diseases, including measles (*Signorelli et al., 2018*; *D'Ancona et al., 2018*; *Italian Ministry of Health, 2017*). Vaccination against measles is now free of charge and mandatory for all children under 16 years. Unvaccinated children are not allowed to attend kindergartens, and financial penalties are imposed on the parents of unvaccinated students attending higher school levels. This regulation has the potential to increase vaccine uptake in new birth cohorts and to immunize school-age children who have escaped routine vaccination (*Trentini et al., 2019*). However, the new policy will not impact the existing immunity gaps in older age groups. In particular, the achievement and maintenance of high vaccination coverage among children may not be enough to avoid the reemergence of measles in the future (*Trentini et al., 2017*; *Trentini et al., 2019*; *Durrheim, 2017*). In order to progress towards measles elimination, it is thus crucial for Italy to identify feasible, sustainable, and effective strategies to reduce the number of susceptible individuals among those who have already left the school system (*Filia et al., 2017*; *Trentini et al., 2017*; *Durrheim, 2017*; *Thompson, 2017*).

The aim of this work is to propose and investigate the effectiveness of a vaccination strategy to be introduced on top of the current policy. The proposed strategy consists of offering vaccination to the parents of all of the children who receive any measles vaccine dose.

## Materials and methods

We simulated the socio-demographic structure of the Italian population over the 2017–2045 period, using an individual-based model of household generation and taking advantage of projections on the future evolution of the age distribution of the Italian population, as provided by the Italian National Institute of Statistics (ISTAT) (see *Supplementary file 1*) (*Italian National Institute of Statistics, 2018*; *Billari et al., 2012*). In the model, individuals are grouped into households following a heuristic approach similar to those previously introduced in the literature (see Appendix 1) (*Fumanelli et al., 2012*; *Marziano et al., 2017*).

The epidemiological status of the population is initialized at the beginning of 2017 using 100 stochastic realizations of the age-specific measles immunity profile estimated for Italy (*Figure 1A*) (*Trentini et al., 2017*). Measles vaccination between 2017 and 2045 is simulated by mimicking vaccination activities carried out during each year, taking into account the age and immunological status of each individual and keeping track of the vaccination history of the individual themselves and of her/his household members.

Two vaccination programs are simulated. The first vaccination program, referred to as the 'current' program, corresponds to the vaccination policy currently in place in Italy, which consists of the routine vaccination of children at 15 months of age, the administration of a second booster dose at 5 years of age, and the check for compliance with this two-dose schedule at both pre-primary and primary school entry. Specifically, as a consequence of the 2017 regulation, children must have received one dose when entering pre-primary schools (at about 3 years of age) and two doses when entering primary schools (at about 6 years of age). The operating guidelines provided by the Ministry of Health also indicate the implementation during the transitional year 2017 of a catch-up campaign targeting all individuals below the age of 16 years who were not compliant with the two-dose schedule (*D'Ancona et al., 2018*; *Italian Ministry of Health, 2018*). Accordingly, in the model, routine vaccination with two-doses is performed every year while a catch-up campaign is simulated in 2017. In addition, from 2018 onwards, measles vaccine is annually offered at pre-primary and primary school entry (i.e., at 3 and 6 years of age) to all children who are not compliant with the routine schedule. Coverage levels for the first and second doses of routine vaccination are assumed to be constant over time and set equal to the most recent estimates of measles vaccination coverage at the national level: 85% and 83%, respectively (*World Health Organization, 2016*). In our simulation, the first dose is administered to children who have never been vaccinated and the second dose is administered to those who have only received one dose. We assume the same vaccination coverage for the 2017 catch-up campaign and for vaccination at school entry. In particular, for these vaccination activities, we assume a baseline coverage level of 50%, which corresponds to current estimates of the impact of the new regulation on measles vaccine uptake in the country (*Italian Ministry of Health, 2019*). Specifically, the coverage level at pre-primary school entry represents the percentage of vaccine uptake among 3-year-old children who have never been vaccinated, whereas the coverage at primary school entry represents the percentage of vaccine uptake among children who have received fewer than two doses.

A second vaccination program, referred to as 'parental vaccination', consists of the implementation from 2018 onwards of a novel strategy targeting the parents of vaccinated children on top of the current program. This supplementary immunization consists of offering measles vaccination to the parents of children who are receiving any dose of measles vaccine. In our simulation, parental vaccination is offered only once to each household, the first time the parents bring one of their children to be vaccinated under the current policy. In particular, we evaluate the impact of parental vaccination under three different coverage scenarios: 50%, 75% and 99%. These percentages represent the proportion of parents who are vaccinated as part of this strategy among all eligible parents, whose exact number depends on the coverage achieved in childhood vaccination programs. We assume that a single vaccine dose is offered to each parent during parental vaccination. In all the considered scenarios, measles vaccine efficacy is set at 95% (*De Serres et al., 1995*).

The effectiveness of each vaccination program is evaluated in terms of its impact on the overall and age-specific susceptibility to infection of the Italian population, on the effective reproduction number $R_e$ over the period 2017–2045, and on the amount of time required to achieve measles elimination. The effective reproductive number $R_e$ represents the expected number of secondary cases generated by one typical infected individual in a partially immune population, where the immunity

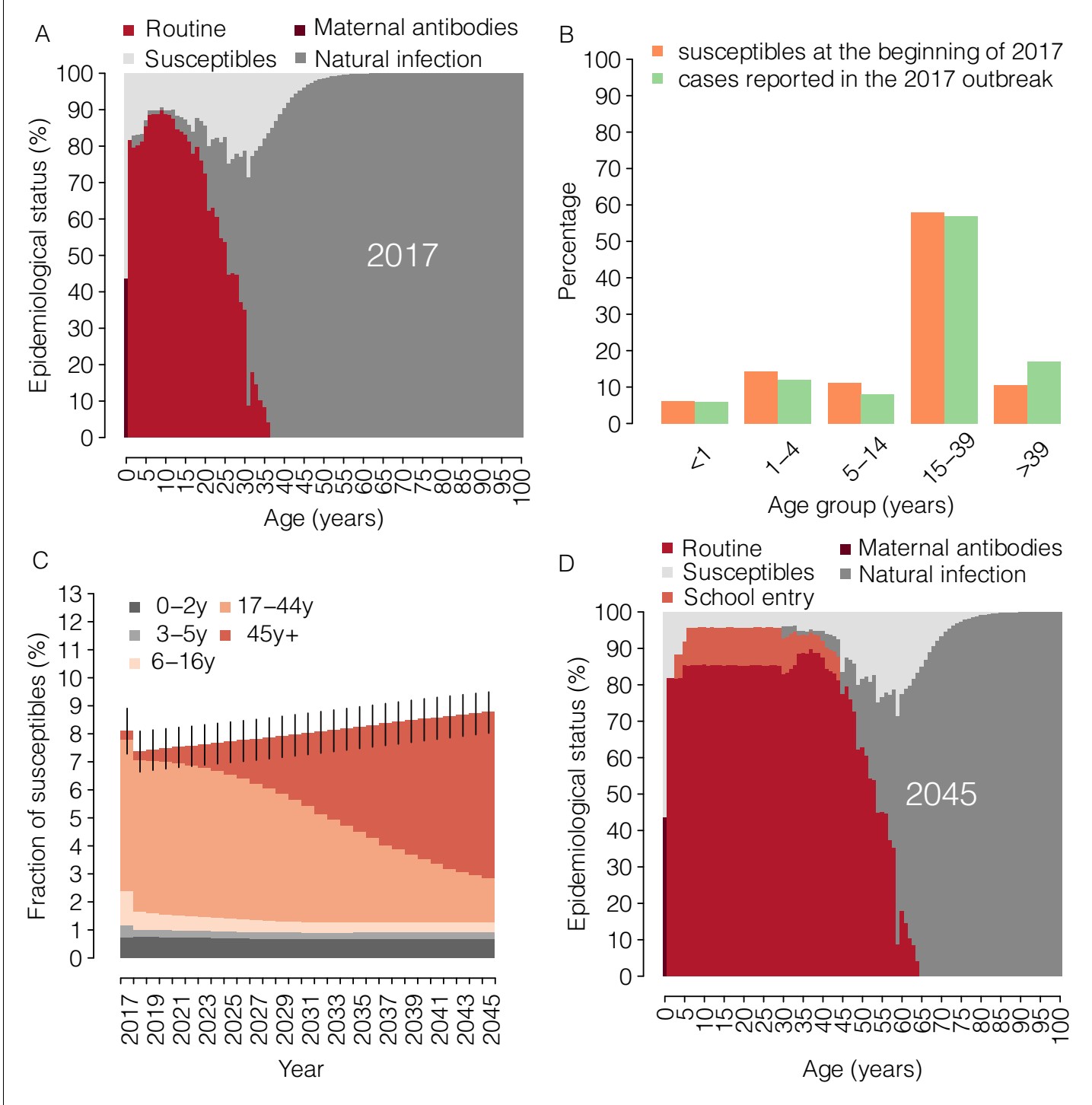

**Figure 1.** Measles epidemiology under the current program (2017–2045). (**A**) Mean measles age-specific epidemiological status as estimated by *Trentini et al. (2017)* at the beginning of the year 2017. Shown for each age is the percentage of individuals who are susceptible to infection or protected against infection by immunity provided by maternal antibodies and by immunity acquired through natural infection or routine (first or second dose) vaccination. (**B**) The age distribution of susceptible individuals at the beginning of 2017, as simulated in our model (orange), and the age distribution of suspected measles cases reported during 2017 to the National Measles and Rubella Integrated Surveillance System (green) (*Italian National Institute of Health, 2017*). (**C**) Mean yearly fraction of susceptible individuals in Italy as estimated by the model for the period 2017–2045 under the 'current' vaccination program. Different colors correspond to different age groups; vertical bars represent 95% confidence intervals (CI) of model simulations. (**D**) Mean measles age-specific epidemiological status as obtained by the model for 2045 under the 'current' vaccination program.

*Figure 1 continued on next page*

*Figure 1 continued*

Shown for each age is the percentage of individuals who are susceptible to infection or protected against infection by maternal antibodies and by immunity acquired through natural infection, routine (first or second dose) vaccination or vaccination at school entry.

DOI: https://doi.org/10.7554/eLife.44942.003

The following source data is available for figure 1:

**Source data 1.** Source data of *Figure 1A*.

DOI: https://doi.org/10.7554/eLife.44942.004

**Source data 2.** Source data of *Figure 1B*.

DOI: https://doi.org/10.7554/eLife.44942.005

**Source data 3.** Source data of *Figure 1C*.

DOI: https://doi.org/10.7554/eLife.44942.006

**Source data 4.** Source data of *Figure 1D*.

DOI: https://doi.org/10.7554/eLife.44942.007

within the population may be due to either vaccination or natural infection. $R_e$ provides important indications of the transmission potential of the virus in the population. If $R_e > 1$ the infection may spread in the population; otherwise, the infection will die out. The year of measles elimination is here defined as the first year between 2017 and 2045, in which $R_e$ falls below 1.

Estimates of $R_e$ between 2017 and 2045 are obtained as follows:

i. we estimate the exponential growth rate *r* of the 2017 measles epidemic, by fitting a linear model to the logarithm of the weekly cases reported to the Italian National Institute of Health (*Italian National Institute of Health, 2017*; *Chowell et al., 2004*; *Wallinga and Lipsitch, 2007*);

ii. we assume that the measles transmission dynamics follow a susceptible-latent-infectious-removed (SLIR) model and we adopt the Wallinga and Lipsitch approach (*Wallinga and Lipsitch, 2007*) to estimate $R_e$ in 2017 as $R_e^{2017} = \frac{1}{\left(\frac{\omega}{\omega+r}\right)\left(\frac{\gamma}{\gamma+r}\right)}$, where $1/\omega = 6.5$ days is the average latent period and $1/\gamma = 7.5$ days is the average infectious period, therefore considering an average generation time of 14 days (*Anderson and May, 1991*);

iii. for each vaccination scenario and each year *y* between 2018 and 2045, we estimate the effective reproduction number $R_e(y)$ as the spectral radius of the next generation matrix encompassing information on the age-specific immunity levels resulting from vaccination and time varying demography, and the age-specific mixing patterns estimated for Italy (see Appendix 1) (*Mossong et al., 2008*; *Diekmann et al., 1990*; *Diekmann et al., 2010*).

The results presented in this paper are based on 1000 different model realizations for each vaccination scenario and include uncertainty regarding: the demographic projections of the age structure of the Italian population over the 2018–2045 period (*Italian National Institute of Statistics, 2018*); the age-specific measles immunity profiles estimated for Italy for 2017 (*Trentini et al., 2017*); the estimated growth rate *r* of the 2017 measles epidemic; and the age-specific mixing patterns of the Italian population (*Mossong et al., 2008*). Details are reported in Appendix 1 and all data required by our simulations are provided as Supplementary Files.

## Sensitivity analysis

We perform different sensitivity analyses to assess the robustness of the obtained estimates when considering:

i. higher coverage for measles vaccination at school entry (75% and 99%, instead of 50% as assumed in the baseline analysis);

ii. shorter/longer generation time for measles (10 and 18 days, instead of 14 days as assumed in the baseline analysis);

iii. different assumptions on population mixing, including an alternative contact matrix estimated for Italy through a modeling approach (*Fumanelli et al., 2012*) and an homogeneous mixing in the population;

iv. an alternative measles transmission model accounting for two distinct phases of infectivity.

In Appendix 1, we also report the results obtained when measles epidemiology is simulated by considering the vaccination strategy adopted in Italy before the introduction of mandatory

vaccination at school entry in July 2017, and present a sensitivity analysis to assess the robustness of the estimates of the exponential growth rate $r$ associated with the 2017 epidemic when including possible underreporting of cases (*Ciofi Degli Atti et al., 2002*).

## Results

From the analysis of measles cases reported during the 2017 outbreak, we estimated an effective reproduction number of 1.66 (95% CI 1.55–1.76). According to our simulations based on the estimates provided by *Trentini et al. (2017)* at the beginning of 2017, 8.1% (95% CI 7.3–8.9) of the Italian population was susceptible to measles. About one third of the susceptible individuals were younger than 16 years, whereas 60% of them were aged between 18 and 45 years (*Figure 1A*). We estimate that the number of measles cases reported during the 2017 Italian outbreak represented only 0.1% (95% CI 0.09–0.12) of the susceptible population in Italy. This implies that, in Italy, about 4.9 million (95% CI 4.4–5.4) people may still be susceptible to measles infection. The age distribution of measles susceptible individuals matches the fraction of cases by age group reported during the 2017 outbreak (*Figure 1B*), thereby confirming the reliability of simulated immunity gaps in the population (*Durrheim, 2016*). According to our results, the catch-up campaign implemented in 2017 under the current program has contributed the immunization of 445189 (95% CI: 394797–487621) susceptible children under 16 years of age, producing a 9% reduction in the overall number of susceptible individuals in 2018 (*Figure 1C*). However, the obtained results show that after this initial drop, the overall fraction of susceptibles would progressively increase in the next decades, reaching 8.8% (95% CI 8.1–9.5) in 2045 (*Figure 1C*). This increase is ascribable to the replacement of elderly individuals, who are predominantly immune because of natural infection, with new birth cohorts that have been only partially immunized as a consequence of suboptimal coverage (*Figure 1A and D*). In particular, we estimate that in 2045 only 14.3% (95% CI: 12.5–15.9) of the susceptible population would be younger than 17 years, while individuals aged more than 45 years, who currently contribute only marginally to the residual measles susceptibility, would represent 69.6% (95% CI: 67.8–71.6) of the total number of susceptibles. As expected, the estimated percentage of susceptibles among individuals currently aged between 17 and 44 years would not be affected by this policy, remaining 16.2% (95% CI 14.5–17.7) in 2045 (*Figure 1D*).

The introduction of parental vaccination on top of the current program has the potential to progressively reduce the immunity gaps in adults as well as the overall susceptibility of the Italian population (*Figures 2* and *3*). Remarkably, the estimated total fraction of susceptible individuals in 2045 under the parental vaccination program ranges between 6.3% (95% CI: 5.8–6.9) for 99% vaccination coverage of parents and 7.6% (95% CI: 6.9–8.2) for 50% vaccination coverage, instead of 8.8% expected under the current program (*Figure 3*). This strategy targets age groups that would otherwise never be reached by the current immunization program, that is cohorts of individuals older than 16 years in the year 2017. In particular, by 2045, parental vaccination at 50% of coverage would result in a 17.1% (95% CI: 16.8–17.5) reduction in the number of susceptible individuals aged between 17 and 44 years in 2017, while a 35% (95% CI: 34.1–35.9) reduction in this age group is expected if 99% coverage is assumed.

According to our analysis, the current program would decrease the measles effective reproduction number to 1.08 (95% CI 0.95–1.23) in 2045 (*Figure 4*). In our simulations, measles elimination is achieved before 2045 only in 12.0% of model realizations.

The introduction of parental vaccination could accelerate the progress towards measles elimination, although the effectiveness of this strategy would depend on parents' response to the new policy. Our simulations show that the effective reproduction number in 2045 would be 0.95 (95% CI 0.82–1.11), 0.89 (95% CI 0.77–1.07), 0.84 (95% CI 0.71–1.04) when 50%, 75% and 99% of eligible families accept parental vaccination, and that with these levels of vaccination acceptance measles elimination would be achieved on average in 2042, 2037 and 2031, respectively. Our results clearly show that parental vaccination has the potential to reduce the risk of major measles epidemics dramatically in the coming decades, although it is difficult to forecast the probability that measles outbreaks will be experienced in the future (details can be found in Appendix 1).

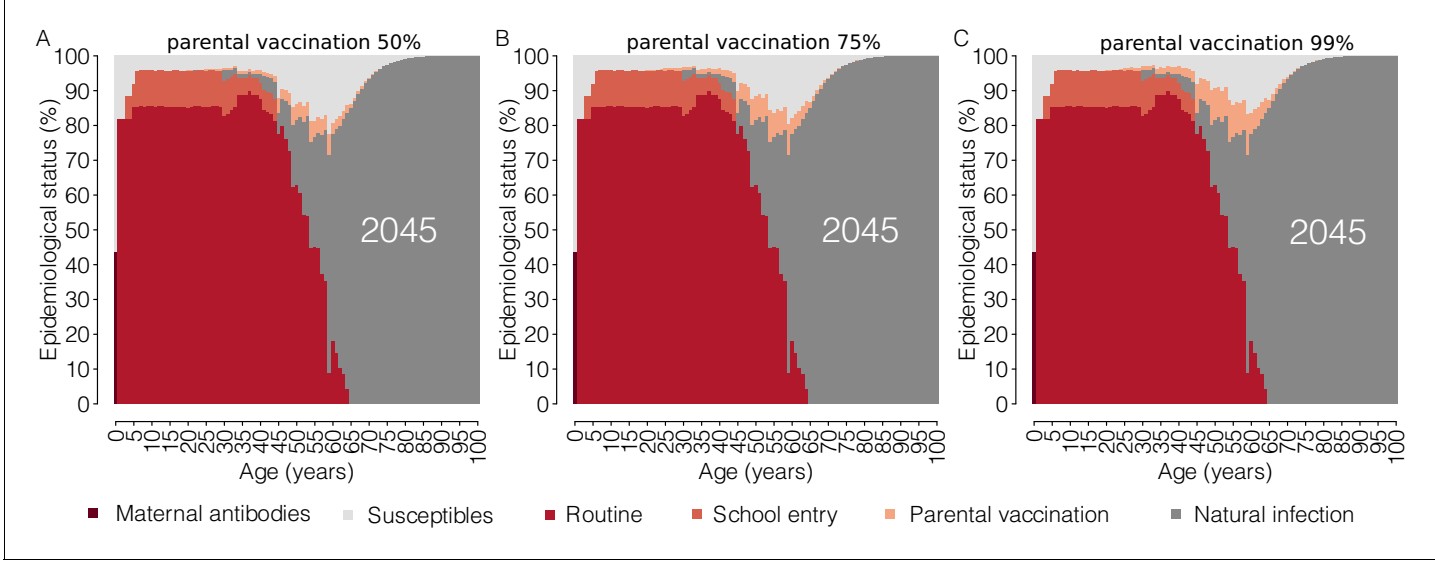

**Figure 2.** Impact of parental vaccination on the future age-specific immunity profiles. Mean measles age-specific epidemiological status as estimated by the model for the year 2045 under different scenarios for the 'parental vaccination' program. Shown for each age is the percentage of individuals who are susceptible to infection or protected against infection by maternal antibodies and by immunity acquired through natural infection, routine (first or second dose) vaccination, vaccination at school entry or parental vaccination.

DOI: https://doi.org/10.7554/eLife.44942.008

The following source data is available for figure 2:

**Source data 1.** Source data of *Figure 2A*.
DOI: https://doi.org/10.7554/eLife.44942.009
**Source data 2.** Source data of *Figure 2B*.
DOI: https://doi.org/10.7554/eLife.44942.010
**Source data 3.** Source data of *Figure 2C*.
DOI: https://doi.org/10.7554/eLife.44942.011

## Sensitivity analysis

The performed analysis shows that an improvement of vaccine uptake at school entry to achieve 99% coverage in the current program may anticipate the timing of measles elimination to 2039. If vaccine uptake at school entry were to be 75%, measles elimination could be achieved in 2042, which is comparable to what might be obtained by reaching 50% of eligible families with parental vaccination in the baseline analysis. By contrast, our simulations show that, under the most optimistic scenario of 99% of coverage both for parental vaccination and vaccination at school entry measles elimination could be achieved, on average, as early as 2023.

The assumption of a shorter or longer generation time would affect model estimates of the effective reproduction number over time. In particular, under parental vaccination at 50% of coverage, a shorter (longer) generation time would result in an anticipation (delay) of the timing of measles elimination, which is estimated to occur before 2045 in 99.2% (16.8%) of model realizations. When a generation time lasting 18 days is considered, the current policy at current coverage levels was insufficient to achieve measles elimination by 2045 in 99.9% of model realizations.

The obtained estimates are qualitatively robust when considering alternative age-specific mixing patterns for the Italian population, although the inclusion of contact matrices estimated through the modeling approach (*Fumanelli et al., 2012*) results in delayed measles elimination under all considered vaccination scenarios. On the other hand, under the (hardly realistic) scenario of a population that mixes fully at random (i.e., by assuming homogeneous mixing), neither the current policy nor its combination with parental vaccination would be sufficient to achieve measles elimination by 2045.

Qualitative temporal patterns in the evolution of the effective reproduction number estimated by exploring different levels of measles transmissibility during the prodromal and exanthema phase are generally robust. The largest quantitative difference can be detected when most secondary cases

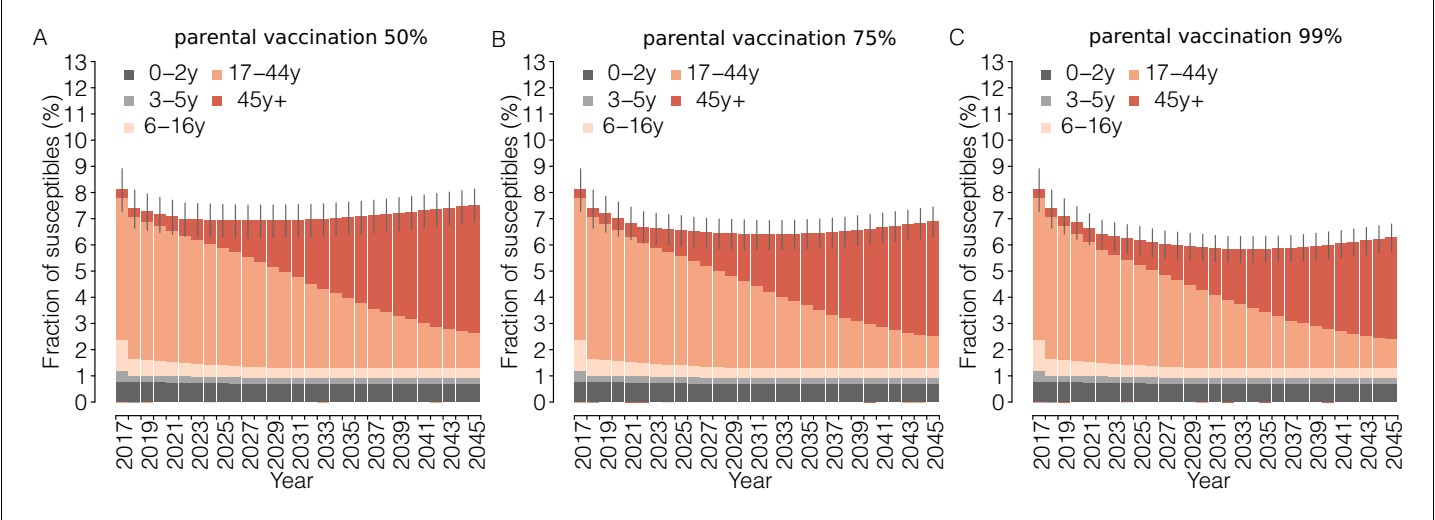

**Figure 3.** Impact of parental vaccination on the proportion of measles-susceptible individuals. Mean yearly fraction of susceptible individuals in the Italian population as estimated by the model for the period 2017–2045 under different scenarios for the 'parental vaccination' program. Different colors correspond to different age groups; vertical bars represent the 95% CI of the model simulations.

DOI: https://doi.org/10.7554/eLife.44942.012

The following source data is available for figure 3:

**Source data 1.** Source data of *Figure 3A*.
DOI: https://doi.org/10.7554/eLife.44942.013
**Source data 2.** Source data of *Figure 3B*.
DOI: https://doi.org/10.7554/eLife.44942.014
**Source data 3.** Source data of *Figure 3C*.
DOI: https://doi.org/10.7554/eLife.44942.015

are generated in the prodromal phase. In this case, under the current policy, measles elimination is predicted to occur before 2045 in 73.9% of model realizations instead of the 12.0% of model realizations seen for the baseline analysis. Similarly, under parental vaccination at 50% of coverage, when most of secondary cases are generated in the prodromal phase, measles elimination is predicted to occur before 2045 in 98.7% of model realizations compared to the 78.9% of model realizations seen in the baseline analysis.

Finally, when considering an extreme scenario in which only 25% of measles cases were reported during the 2017 outbreak, we estimate the exponential growth rate to be 0.29 (95% CI: 0.21–0.37), similar to that obtained when only reported cases are used: 0.29 (95% CI: 0.25–0.33). As estimates of the effective reproduction number depend only on the growth rate and on measles natural history, these results suggest that our findings are robust with respect to the reporting rate (and size) of the 2017 measles outbreak.

Details on the performed sensitivity analyses are reported and discussed in Appendix 1.

## Discussion

In July 2017, the Italian Government approved a regulation requiring parents to vaccinate their children before school entry against ten infections, including measles. Recent estimates suggest that the new regulation allowed the vaccination of 50% of individuals who escaped routine vaccination (*Italian Ministry of Health, 2019*; *Italian National Institute of Health, 2019*). Our modeling study shows that the current policy would reduce measles susceptibility in the age segments of the population characterized by higher contact rates, resulting in a remarkable decrease in the infection transmission potential and making measles elimination a realistic target. However, if only 50% of unvaccinated children are vaccinated at school entry, disease elimination would probably be achieved only after 2045.

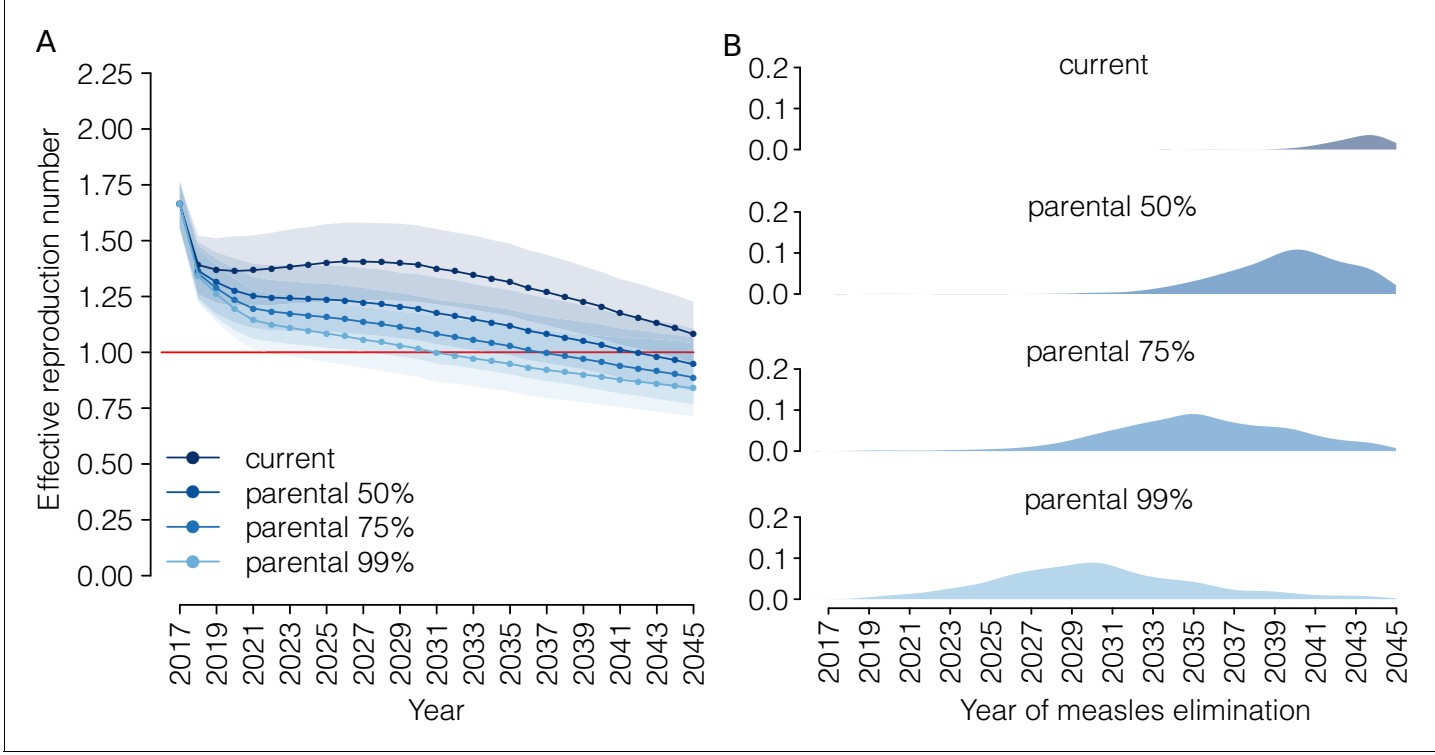

**Figure 4.** Progress towards measles elimination. (**A**) Mean effective reproduction number over time, as estimated by the model under the 'current' vaccination program and under different coverage scenarios for the 'parental vaccination' program. Shaded areas represent the 95% CI associated with model estimates. The red line represents the measles elimination threshold; elimination is achieved when the effective reproductive number is smaller than 1. (**B**) Probability associated with different time at measles elimination, as obtained by 1000 model realizations under the 'current' vaccination program and under different coverage scenarios for the 'parental vaccination' program.

DOI: https://doi.org/10.7554/eLife.44942.016

The following source data is available for figure 4:

**Source data 1.** Source data of *Figure 4A*.
DOI: https://doi.org/10.7554/eLife.44942.017
**Source data 2.** Source data of *Figure 4B*.
DOI: https://doi.org/10.7554/eLife.44942.018

Offering vaccination to the parents of children who receive a measles vaccine dose could progressively reduce by 17–35% the immunity gaps among individuals who are between 18 and 45 years of age in 2018. The implementation of this program would decrease the overall susceptibility of the population by 6.2–22.0%, and would increase the probability of measles elimination before 2045 from 12.0% (estimated in the absence of this additional policy) to 78.9–96.5%. The effectiveness of this strategy clearly depends on both the coverage achieved through childhood immunization (routine programs and vaccination at school entry) and on the willingness of parents to be vaccinated themselves. The obtained estimates are generally robust with respect to different assumptions on the duration of measles generation time and on the relative transmissibility of measles during the prodromal and exanthema phases. On the other hand, under the assumption of homogeneous mixing in the population, neither the current immunization program nor parental vaccination appear to be sufficient to eliminate measles before 2045.

The study presents a few limitations that should be carefully considered in order to achieve a better interpretation of the obtained results. In particular, our estimates of the effective reproduction number were obtained using measles cases reported during the 2017 outbreak. The current degree of measles underreporting in statutory notifications is unknown. However, our estimates of the effective reproduction number are stable with respect to the possible underreporting of cases during the outbreak (*Ciofi Degli Atti et al., 2002*). The proposed analysis did not take into

account potential geographical heterogeneities in measles immunity levels at the sub-national scale. Although the new regulation is expected to harmonize the vaccine offer and its uptake in Italy, significant regional differences in both immunization schedule and coverage have been reported in the past (*Bonanni et al., 2015*). Regions characterized by a lower than national average vaccine uptake in the past may therefore experience a delay in measles elimination with respect to the results presented in this work. In our work, future measles susceptibility might have been overestimated, as we did not explicitly model measles transmission, thus disregarding the impact of future measles spread on the immunity profile of the Italian population. Although both the occurrence and magnitude of future measles epidemics are largely uncertain and difficult to predict (*Earn et al., 2000*), changing patterns of measles transmission may affect both the number of susceptible adults and the incidence of severe disease in the coming years. However, the population infected during the 2017 Italian outbreak—one of the largest occurred in Europe in the last years—represented only 0.1% of the estimated susceptible population in the country (*Italian National Institute of Health, 2017*; *Trentini et al., 2017*). This suggests that the explicit inclusion of measles transmission may have a limited impact on short- or medium-term estimates of the immunity profile and measles transmission potential. The proposed analysis relies on the simplifying assumption that parents decide whether to vaccinate their children regardless of past vaccination behavior, although it is likely that parents vaccinate either all or none of their children. All children receiving vaccination indirectly present their parents with the opportunity to vaccinate themselves. As the children receiving vaccination may be clustered in a smaller number of households than is the case in our model, we are probably overestimating the potential number of parents who are eligible for measles vaccination. In particular, in our simulations, 98.7% of families with children between 1 and 15 years of age are considered as eligible for parental vaccination, whereas in a perfectly clustered model, this percentage would be 88.1%. On the other hand, clustering of unvaccinated children may have a larger effect on measles transmission dynamics than on the number of parents eligible for vaccination. Finally, we assumed that routine vaccination coverage would not be affected by the implementation of the new national policy, and that the coverage of the catch-up campaign conducted in 2017–2018 was the same as that of vaccination at school entry (both for pre-primary and for primary schools). However, data released in December 2018 by the Italian Ministry of Health suggest that the new regulation on mandatory vaccination at school entry may have indirectly affected the first-dose vaccine uptake for children under 3 years of age. In particular, the available records show that the first-dose vaccination coverage in the 2015 age cohort has increased from 91.4% in 2017 to 94.2% in 2018, (*Italian Ministry of Health, 2019*) although the fraction of unvaccinated children who were vaccinated thanks to the new regulation may vary depending on the age cohort considered (*D'Ancona et al., 2018*; *Italian Ministry of Health, 2019*). According to the most recent estimates, measles vaccination coverage in the 2014 age cohort has increased from 87.3% in 2016 to 94.4% in 2018, suggesting that the new regulation resulted in the vaccination of about 56% of unvaccinated children in this cohort (*Italian National Institute of Health, 2019*).

In conclusion, our analysis shows that a marked increase in childhood immunization rates would not be sufficient to achieve measles elimination in the short- or medium-term in Italy. These results confirm the need for appropriate strategies to vaccinate individuals who have already left the school system in order to reduce critical immunity gaps in young adults (*Trentini et al., 2019*; *Filia et al., 2017*; *Trentini et al., 2017*; *Durrheim, 2017*; *Thompson, 2017*; *Wise, 2018*; *Gidding et al., 2007*). Attempts made to date in this direction either have only been partially effective or have required remarkable efforts in terms of the costs to and commitment of the public health authorities (*Gidding et al., 2007*; *Morice et al., 2003*; *Kelly et al., 2007*). In Costa Rica, a measles-rubella vaccination campaign targeting adults aged 15–39 years was successfully conducted in 2000, but it required huge efforts of communication, social mobilization, and the use of house-to-house vaccination teams (*Morice et al., 2003*). In 2001–2002, a vaccination campaign in Australia targeting young adults aged between 18 and 30 years who visited their general practitioner (GP) had little effect on the immunity gaps, probably because of a lack of promotion and central coordination (*Gidding et al., 2007*; *Kelly et al., 2007*). In Europe, beyond some local attempts to immunize adolescents and individuals before school leaving, which have only marginally affected the vaccine uptake (*Lashkari and El Bashir, 2010*; *Vazzoler et al., 2014*), little has been done to reduce residual susceptibility in adults. Interventions recently set up include attempts to raise awareness in people attending social events that may represent potential hotspots for measles transmission

(*Public Health England, 2018*). In this work, a new strategy is proposed, consisting of offering vaccination to parents of children who are being vaccinated against measles. Although the proposed policy can reach only a fraction of susceptible adults, that is those with children in the measles-vaccination age group, the obtained results suggest that this strategy may be both feasible and effective. In particular, our results suggest that vaccinating 50% of parents who agreed to vaccinate their children, and may therefore be inclined towards accepting vaccination, would promote measles elimination as well as reaching 50% of children who still escape measles vaccination despite the fact that vaccination is now mandatory in Italy (i.e., increasing vaccination coverage at school entry from 50% to 75%). The sustainability of the proposed strategy should be carefully evaluated by public health decision makers. However, a key advantage of this policy is that it does not require targeted activities to recruit parents, thus resulting in a relatively simple implementation protocol.

Beyond parental vaccination, alternative immunization strategies aimed at reducing residual susceptibility in adults may also be considered. These may include the extension of mandatory vaccination at university entry – an intervention already implemented in different US states. Other immunization efforts may include the introduction of proof of immunity as a condition for the enrolment of health care workers (HCWs), for whom measles vaccination is only recommended in most European countries (*Galanakis et al., 2014*; *Maltezou et al., 2019*). The need to improve vaccination coverage among HCWs is due to their potential to amplify measles outbreaks and their higher risk of exposure to the virus, as observed in the 2017 Italian outbreak in which 7% of cases were HCWs (*Maltezou et al., 2019*).

The achievement of measles elimination remains a global health priority. Actions may be also required to raise awareness and consensus about the benefits coming from vaccination and to increase the overall vaccine uptake. Country-specific policies should be identified and carefully evaluated by decision makers in order to anticipate the time of measles elimination as much as possible.

## Acknowledgements

We thank Dr. Maria Litvinova for contributing to the validation of the household generation model against census data and for useful comments on the manuscript.

## Additional information

### Funding
The authors declares that there was no funding for this work.

### Author contributions
Valentina Marziano, Conceptualization, Data curation, Software, Formal analysis, Validation, Investigation, Visualization, Methodology, Writing—original draft, Writing—review and editing; Piero Poletti, Conceptualization, Formal analysis, Supervision, Validation, Investigation, Methodology, Writing—review and editing; Filippo Trentini, Data curation, Writing—review and editing; Alessia Melegaro, Writing—review and editing; Marco Ajelli, Supervision, Methodology, Writing—review and editing; Stefano Merler, Conceptualization, Resources, Supervision, Methodology, Project administration, Writing—review and editing

### Author ORCIDs
Valentina Marziano   https://orcid.org/0000-0003-2842-7906
Alessia Melegaro   http://orcid.org/0000-0003-2221-8898

### Decision letter and Author response
Decision letter https://doi.org/10.7554/eLife.44942.038
Author response https://doi.org/10.7554/eLife.44942.039

# Additional files

## Supplementary files

• Source code 1. Code used to run model simulations (C programming language).
DOI: https://doi.org/10.7554/eLife.44942.019

• Supplementary file 1. Stochastic demographic projections of the age structure of the Italian population over the 2018–2045 period as provided by the Italian National Institute of Statistics (ISTAT).
DOI: https://doi.org/10.7554/eLife.44942.020

• Supplementary file 2. Bootstrapped contact matrices as obtained from random sampling of participants to the POLYMOD contact survey.
DOI: https://doi.org/10.7554/eLife.44942.021

• Transparent reporting form
DOI: https://doi.org/10.7554/eLife.44942.022

## Data availability

In this work, no new data were collected. Performed experiments only consist of model simulations, which take advantage of previously published data from different sources. Proper references to these data can be found in the manuscript and in Appendix 1 or they were made available as Supplementary Files.

The following previously published datasets were used:

| Author(s) | Year | Dataset title | Dataset URL | Database and Identifier |
|---|---|---|---|---|
| Mossong J, Hens N, Jit M, Beutels P, Auranen K, Mikolajczyk R, Massari M, Salmaso S, Scalia Tomba G, Wallinga J, Heijne J, Sadkowska-Todys M, Rosinska M, Edmunds WJ | 2017 | POLYMOD social contact data (Version 1.1) | https://zenodo.org/record/1215899#.XTBQeZMzbVo | Zendo, 10.5281/zenodo.1215899 |

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

# Appendix 1

DOI: https://doi.org/10.7554/eLife.44942.023

The following supporting text contains methodological details and additional results.

## 1. Materials and methods

In this study, we investigate the effectiveness of a novel vaccination strategy, to be added on top of the current immunization program, in reducing measles susceptibility in Italy. The proposed strategy, here denoted as 'parental vaccination', consists of offering vaccination to parents who bring their children to be vaccinated against measles. The impact of parental vaccination on measles epidemiology is evaluated using a model that is capable of reproducing not only the age distribution of the Italian population but also the main features of Italian household demography, including the distribution of households' size, the heterogeneity in household composition, and the age gaps between household members. To this aim, an individual-based model of household generation previously introduced in the literature (*Merler et al., 2011*) was adapted to simulate the time-varying socio-demographic structure of the Italian population over the period 2017–2045. The model takes advantage of data on and projections of the Italian age structure, as provided by the Italian National Institute of Statistics (ISTAT), and of the most recent census data on the composition and size of Italian households (*Italian National Institute of Statistics, 2018*; *Statistical Office of the European Commission, 2011*). Projections of the Italian population used in this study (see *Supplementary file 1*) are based on different stochastic realizations of official forecasts as provided by ISTAT and obtained through a method introduced in the literature by Billari and colleagues in 2012 (*Billari et al., 2012*). This method relies on the framework of the so-called 'random-scenario approach', which is based on a series of subsequent expert-based conditional evaluations of the future evolution of different demographic indicators, given the values of the indicators at previous time points. Component-specific forecasts are combined and applied to an initial population (2017) to obtain different projections of the age-structure and overall size of the Italian population between 2018 and 2060 (*Billari et al., 2012*; *Italian National Institute of Statistics, 2019*).

### 1.1 Model of household generation

For the year 2017, the proposed model simulates a population of households of different sizes (of between 1 and 7 household members) in such a way as to reproduce a number of individuals that matches the observed Italian population size (~60 million) and to mirror the observed distribution of households by size and type (e.g., couple with children, single adult, etc.) (*Statistical Office of the European Commission, 2011*). For each simulated household, the age of the household members is assigned according to the observed age distribution by household size (*Italian National Institute of Statistics, 2018*; *Statistical Office of the European Commission, 2011*).

The age of the individuals represented in the model faithfully reproduces the age structure of the Italian population observed in 2017 (*Appendix 1—figure 1a*). Moreover, the households generated by the model well match the observed distribution of Italian households by size and by type (*Appendix 1—figure 1b-c*). In particular, according to the model and to census data, more than 70% of Italian households have fewer than four members and the great majority of households of size three or more (~80%) are couples with children. The model is also capable of reproducing the observed age distribution of members in the different household sizes (*Appendix 1—figure 1d-l*).

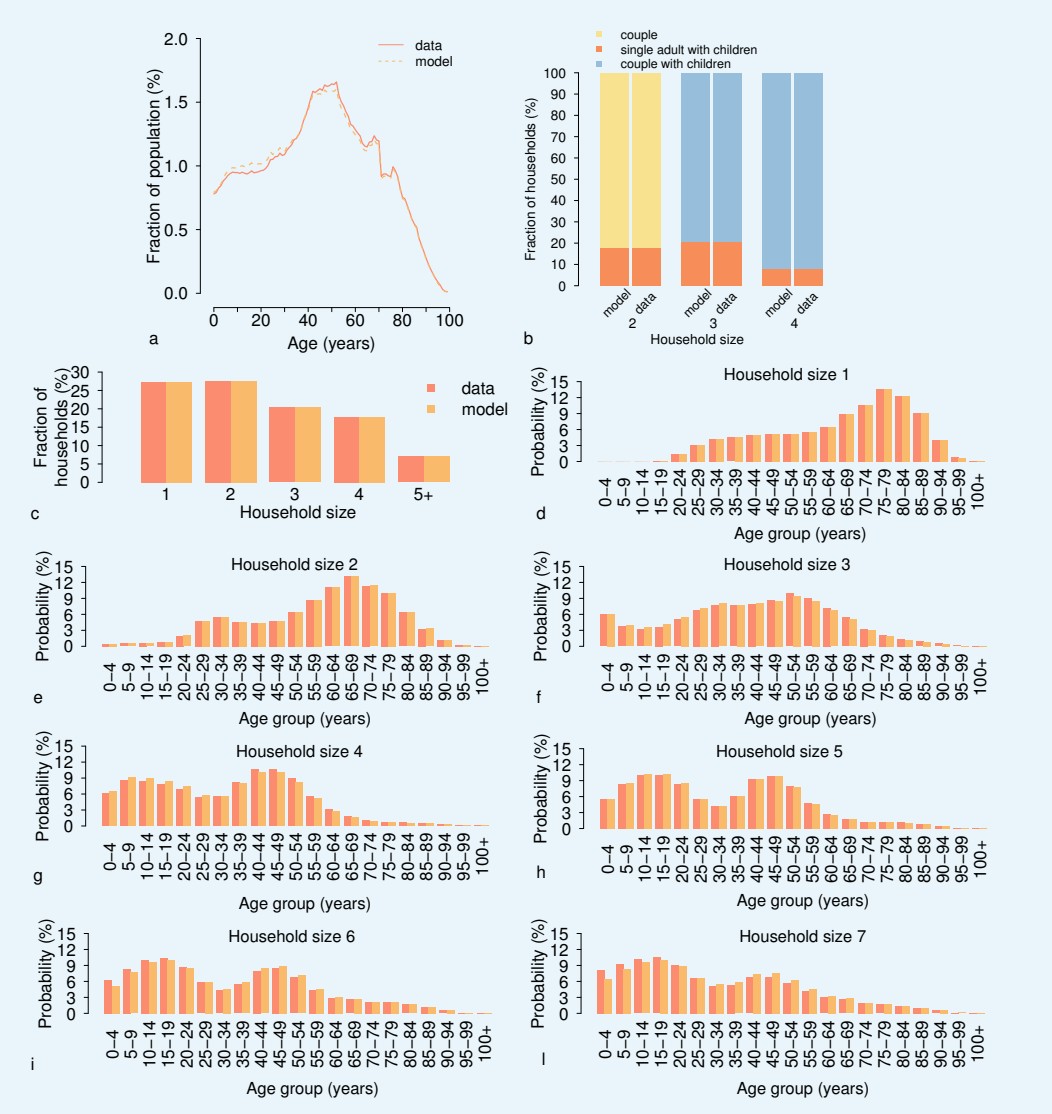

**Appendix 1—figure 1.** Age distribution and household structure of the Italian population in the year 2017. (**a**) Age distribution of the Italian population as simulated by the model (dotted line) and as reported by the Italian National Institute of Statistics (solid line) for 2017 (*Italian National Institute of Statistics, 2018*). (**b**) Fraction of households of different type (%) as simulated by the model and as observed in the census data (*Statistical Office of the European Commission, 2011*) for different household sizes. (**c**) Fraction of households by size (%) as simulated by the model and as observed in census data (*Statistical Office of the European Commission, 2011*). (**d-l**) Age distribution of household members by household size h∈{1,. . .,7} as simulated by the model and as observed in census data (*Statistical Office of the European Commission, 2011*).

DOI: https://doi.org/10.7554/eLife.44942.024

As for the age difference between household members, we compared a set of statistics based on the last census data to the corresponding data set obtained from our simulations (*Appendix 1—figure 2*). Although with some limitations, the model satisfactory complies with a set of micro-level data: the age distributions of the oldest parent's age disaggregated by age of the youngest one (*Appendix 1—figure 2a-e*), age gaps between parents and their children (*Appendix 1—figure 2f*), and age difference between the youngest of the parents and his/her children (*Appendix 1—figure 2g*).

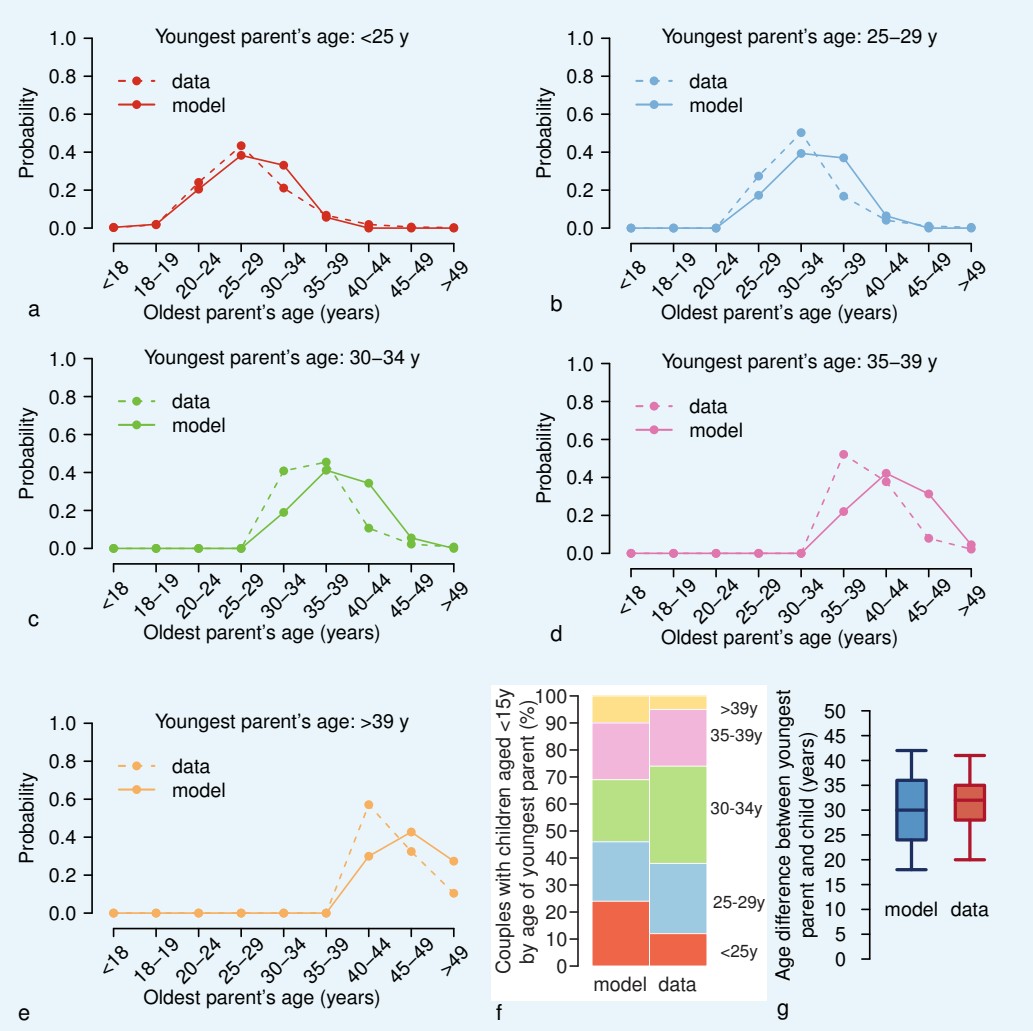

**Appendix 1—figure 2.** Age differences between household members. (**a**) Age distribution of the oldest parent in households in which the youngest parent is less than 25 years old as simulated by the model (solid line) and as observed in the census data (dotted line) (***Statistical Office of the European Commission, 2011***). (**b-e**) As (**a**) but for households in which the youngest parent is aged (**b**) 25–29 years, (**c**) 30–34 years, (**d**) 35–39 years, and (**e**) > 39 years. (**f**) Simulated and observed fractions of couples (%) with children younger than 15 years disaggregated by age of the youngest parent. (**g**) Boxplots (percentiles: 2.5, 25, 50, 75, and 97.5) of the age difference between the youngest parents and their children as simulated by the model (blue) and as observed in the census data (red) (***Statistical Office of the European Commission, 2011***).

DOI: https://doi.org/10.7554/eLife.44942.025

By taking advantage of the projections of the yearly age distribution of the Italian population (***Italian National Institute of Statistics, 2018***), we simulate the Italian household demography over the period 2018–2045, by generating households and individuals in a manner similar to that used for the year 2017. Specifically, in the absence of information on the future evolution of the structure of Italian households, we assume that the age-specific probability of being part of a household of a certain size remains constant over the period 2018–2045, and equal to that estimated for 2017 from the census data (***Statistical Office of the European Commission, 2011***). Under this assumption, temporal changes in the distribution of households by size result from changes in the age distribution of the Italian population estimated by the available demographic projections (***Appendix 1—figure 3***).

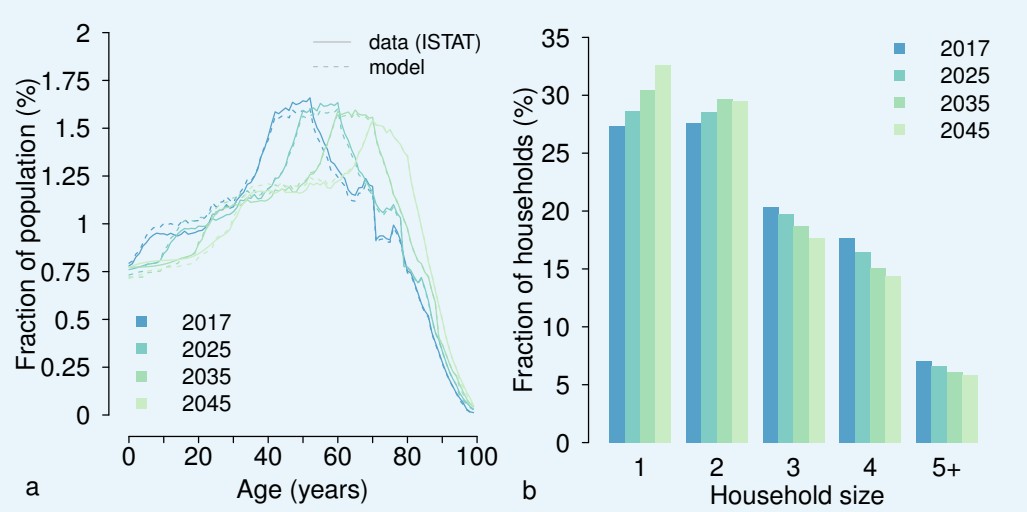

**Appendix 1—figure 3.** Age distribution of the Italian population and household size over the period 2017–2045. (**a**) Age distribution of the Italian population in different years as estimated by the model (dotted lines) and according to the median demographic projections provided by the Italian National Institute of Statistics (*Supplementary file 1*) (*Italian National Institute of Statistics, 2018*). (**b**) Fraction of households by size (%) as obtained by the model for different years.

DOI: https://doi.org/10.7554/eLife.44942.026

The model is capable of reproducing the evolution of the (projected) age distribution of the Italian population throughout the prediction period (*Appendix 1—figure 3a*) (*Italian National Institute of Statistics, 2018*). In our simulations, the population aging expected in the next decades would result in a progressive shift of the distribution of households towards smaller sizes (see *Appendix 1—figure 3b*).

Each individual in the (synthetic) population is defined by the following characteristics:

- household membership;
- age, which is between 0 and 100 years;
- epidemiological status, namely susceptible or immune. For immune individuals, we distinguish whether the individual is immune because she/he has recovered from a natural infection, because of vaccination, or because she/he is temporarily protected by maternal antibodies (~57% of individuals younger than 1 year of age [*Trentini et al., 2017*]); the number of vaccine doses received by the individual is also recorded.

At the beginning of each considered year, a new population of households and individuals is generated, following the algorithm described above. The epidemiological status of each individual is initially assigned on the basis of age-specific proportions obtained at the end of the previous year, and it is updated during the year by mimicking the considered vaccination program/s. In the model, individuals' epidemiological status for the year 2017 is initialized on the basis of 100 stochastic realizations of the measles age-specific immunity profile estimated by *Trentini et al. (2017)* for Italy.

## 1.2 Measles vaccination

Different vaccination programs are simulated for the period 2017–2045 as described hereafter.

## Current program

This program corresponds to the vaccination policy in place in Italy after the approval of regulation 119/2017. According to this program, measles vaccination is mandatory for all minors up to 16 years (*D'Ancona et al., 2018*). In our model, we mimic the immunization activities carried out in 2017 and those planned from 2018 onwards. Specifically:

- For each year from 2017 to 2045, we implement routine vaccination based on a two-dose schedule. A fraction $c_1$ of unvaccinated individuals aged 15 months is vaccinated with a first dose of measles vaccine. A fraction $c_2$ of 5-year-old individuals who received the first dose of the measles vaccine is vaccinated with the second dose. Coverage levels for the two doses of routine vaccination were set at 85% and 83%, respectively (**World Health Organization, 2016**), and kept fixed over the period 2017–2045.

- In the year 2017 only, we implement a catch-up campaign for all minors from 3 years of age up to 16 years. In fact, according to the operational guidelines provided by the Italian Ministry of Health, children below 6 years of age should have received the first dose of measles vaccine, whereas children older than 6 years of age should have received two doses (**Italian Ministry of Health, 2018**). In our model, a first dose is offered to unvaccinated children aged 3–5 years, a second dose is offered to children aged 6–16 years who have already received the first dose, and two doses are offered to children aged 6–16 years who have never been vaccinated. The coverage of this immunization activity, $c_S$, is assumed to be equal across different ages.

- From the year 2018 onwards, the compliance with the two-dose schedule is checked only at school entry. Specifically, each year a fraction $c_{SE}$ of unvaccinated individuals are vaccinated with the first dose at the beginning of pre-primary school (3 years of age) and a fraction $c_{SE}$ of individuals that have missed the second (or both doses) is vaccinated by the entry at primary school (6 years of age). The coverage level for these activities is assumed to be the same as that in the catch-up campaign (i.e. $c_{SE=}c_S$). In our baseline analysis, on the basis of recent estimates of the impact of the new regulation on the measles vaccine uptake in Italy (**Italian Ministry of Health, 2019**), $c_S$ is set at 50%.

## Parental vaccination

In this program, we implement the vaccination of the parents of children targeted by the immunization activities foreseen under the current program. Specifically, for each year between 2017 and 2045, childhood vaccinations are implemented as in the current program. In addition, starting from 2018, measles vaccination is offered to the parents of children who are vaccinated with either their first or second dose of measles vaccine. Vaccination is assumed to be offered to parents only once in life; moreover, we assume that, if two parents are present in the household of the child, either both or neither of them accept vaccination.

The coverage level for parental vaccination is denoted by $c_P$ and indicates the fraction of parents who accept the invitation to be vaccinated among those having children that are vaccinated in a given year. The fraction of parents reached by this program also depends on the coverage assumed for the first and second dose of routine vaccination ($c_1$, $c_2$) and for vaccination at school entry ($c_S$). In our baseline analysis, coverage levels for the first and second dose of routine vaccination and for vaccination at school entry are assumed to be the same as those in the current program. Three different coverage levels for parental vaccination $c_P$ are considered: 50%, 75%, and 99%. Different coverage levels for vaccination at school entry have been explored for sensitivity analysis.

For all the considered programs, vaccine efficacy at each dose administration is set at 95% (**De Serres et al., 1995**), so that approximately 99% of individuals who received two doses are successfully immunized against measles. Specifically, after the administration of a dose, susceptible individuals have 95% probability of becoming immune against measles infection; otherwise their immunological status remains unchanged. The model keeps track at the individual level of failure events, the program under which a vaccine dose is administered (either routine, school entry, or parental vaccination), and (if any) the program under which individuals acquire protection against measles infection.

## 1.3 Estimation of the effective reproduction number

The effective reproduction number $R_e$ represents the average number of secondary cases generated by a typical infected individual in a partly immunized population. This

epidemiological parameter is crucial because it provides information about the transmissibility of the virus in a given population. If the effective reproduction number is lower than 1, the epidemic is expected to go extinct; if it is greater than 1, the infection will have the chance to spread in a population.

In our study, we assume that measles transmission can be represented as a susceptible-latent-infectious-removed (SLIR) model. Briefly, in a SLIR model, susceptible individuals can become infected after a contact with an infectious individual, becoming infectious after an average latent period $1/\omega$. Infectious individuals recover from measles infection after an average time $1/\gamma$ (the infectious period), gaining life-long protection against measles infection. Immune individuals are those individuals who are immune to the infection either because they recovered from natural infection or because they were immunized through successful vaccine administration. For this model, for each vaccination scenario and for each year of interest $y$, the effective reproduction number $R_e(y)$ can be computed as the spectral radius of the next generation matrix $K(y)$ (*Diekmann et al., 1990*; *Diekmann et al., 2010*), where the entry $k_{ij}(y)$ of the matrix is defined as:

$$k_{ij}(y) = q C_{ij} s_j(y)$$

and

- $q$ is an unknown scale factor, shaping the probability of measles transmission given an infectious contact;
- $C_{ij}$ is the average number of contacts that an individual of age $i$ has with individuals of age $j$, according to contact matrices estimated for Italy (*Mossong et al., 2008*);
- $s_j(y)$ represents the fraction of individuals of age $j$ that, according to our model estimates, are susceptible to measles in the year $y$ under the considered vaccination scenario.

As the value of $q$ is unknown, we estimated the effective reproduction number in 2017 by using an approach that is based on the growth rate of the epidemic instead of requiring direct estimates of the transmission rate (*Chowell et al., 2004*; *Wallinga and Lipsitch, 2007*; *Mills et al., 2004*). In particular, we estimated the exponential growth rate $r$ associated with the measles outbreak that recently occurred in Italy, by fitting a linear model to the logarithm of the cases reported to the Italian National Health Institute (*Italian National Institute of Health, 2017*) during the first 5 weeks of 2017 (see *Appendix 1—figure 4*). By assuming that measles transmission follows a SLIR scheme, and using the arguments proposed in *Chowell et al. (2004)*, *Wallinga and Lipsitch (2007)*, and *Mills et al. (2004)*, we calculated the effective reproduction number as:

$$R_e^{2017} = \frac{1}{\left(\frac{\omega}{\omega+r}\right)\left(\frac{\gamma}{\gamma+r}\right)} \tag{1}$$

where, $1/\omega$ and $1/\gamma$ are set to 6.5 and 7.5 days, respectively, thus considering an average generation time of 14 days (*Anderson and May, 1991*).

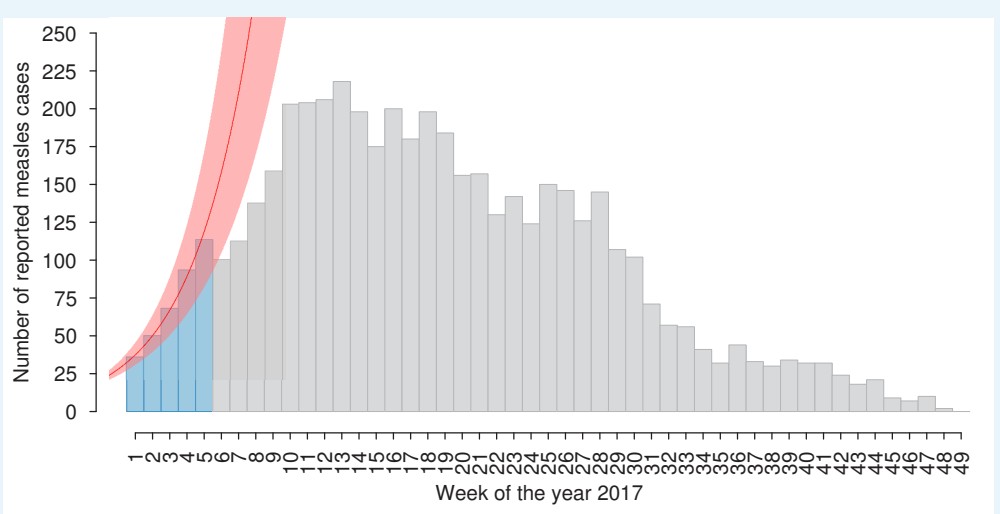

**Appendix 1—figure 4.** Number of new suspected measles cases per week as reported to the Italian National Health Institute in the year 2017 (light-blue and grey bars) (*Italian National Institute of Health, 2017*). The exponential fit (mean and 95% CI) associated with the exponential growth rate r is shown in red. This curve was obtained by fitting a linear model to the logarithm of the number of weekly reported cases in the early epidemic phase (light-blue bars), when the fraction of susceptible individuals in the population is sufficient to sustain exponential growth (*Chowell et al., 2004*; *Wallinga and Lipsitch, 2007*; *Mills et al., 2004*). The time window of 5 weeks represents the largest duration during which the fit provided a coefficient of determination (R-square) larger than 0.99.
DOI: https://doi.org/10.7554/eLife.44942.027

Then, by setting $R_e$ (y=2017)$= R_e^{2017}$, we inferred the value of q.

As the probability of measles transmission given an infectious contact is mainly a biological feature, we can assume that the value of q will remain unchanged in the future. We thus estimate the future evolution of the effective reproduction number $R_e(y)$ by applying the next generation matrix approach to a time varying K, where q is constant and $s_j$ changes as a consequence of the different vaccination policies being considered.

Estimates of $R_e(y)$ presented in this paper include uncertainty regarding:

- the demographic projections of the age structure of the Italian population over 2018–2045;
- the age-specific measles immunity profile estimated for Italy for 2017;
- the estimate of the growth rate r associated with the 2017 measles epidemic;
- the contact matrices $C_{ij}$ estimated for Italy (*Mossong et al., 2008*). Specifically, estimates of $C_{ij}$ were derived by using the publicly available individual contact diaries collected by the POLYMOD study for Italy to generate 1000 bootstrapped contact matrices (provided as *Supplementary file 2*) (*Mossong et al., 2017*).

Specifically, different matrices and time varying trajectories of the age-specific susceptibility, as obtained by including different initial susceptibility for the period 2017–2045 and different demographic projections, were combined with different values of the exponential growth rate r to obtain 1000 simulations incorporating the different levels of uncertainty.

## 1.4 Outbreak probability

An outbreak occurs when the number of cases escalates. According to the mathematical theory of stochastic epidemic models, when disease transmission is non-endemic and $R_e > 1$, the probability of experiencing an outbreak after a single reintroduction of the infection can be computed as:

$$P = 1 - \left(\frac{1}{R_e}\right)$$

where $\left(\frac{1}{R_e}\right)$ represents the probability of disease extinction (**Allen, 2008**).

This equation holds for stochastic SIR epidemic models and can be generalized as

$$P = 1 - \left(\frac{1}{R_e}\right)^n$$

to obtain the probability of outbreak after $n$ reintroductions of the infection, whether reintroductions occur as separated events or not.

This equation can be derived by considering a simple random walk model, in which the random variable $X(t)$ takes values in the set $\{0, 1, 2, \ldots\}$, $p$ is the probability of moving from $x$ to $x+1$, and $q$ is the probability of moving from $x$ to $x-1$. In this model, 0 is an absorbing state and it can be shown (**Allen, 2003**) that the probability of absorption given $p$, $q$ and the initial condition $x_0$ is

$$\lim_{t \to \infty} \text{Prob}\{X(t) = 0\} = \begin{cases} 1, & p \leq q \\ \left(\frac{q}{p}\right)^{x_0}, & p > q \end{cases}$$

In order to compute the probability of outbreak, we can define the random variable $X(t)$ as the number of infected individuals in the population. In a SIR model, after a single reintroduction of the infection, the rate at which new infected individuals are generated can be approximated as $\beta i\, s_o$, where $\beta$ is the transmission rate, $i$ is the number of infected individuals in the population at time $t$, and $s_o$ is the fraction of susceptibles in the population. On the other hand, the rate at which a new recovery occurs is $\gamma i$, where $\gamma$ is the recovery rate. For values of $R_e > 1$, the probability of disease extinction (i.e., absorption) can be written as $\left(\frac{q}{p}\right)^{x_0} = \left(\frac{\gamma i}{\beta i s_o}\right)^n = \left(\frac{1}{R_e}\right)^n$, where $n$ is the initial number of infected individuals.

As a consequence the probability of observing an outbreak can be computed as:

$$\text{Outbreak probability} = \begin{cases} 0, & Re \leq 1 \\ 1 - \left(\frac{1}{R_e}\right)^n, & Re > 1 \end{cases}$$

In **Appendix 1—figure 5**, we show the outbreak probability associated with the vaccination programs discussed in the main text under different assumptions relating to the number of imported cases over a year ($n$). Obtained results clearly show that the outbreak probability can be dramatically different for different values of $n$.

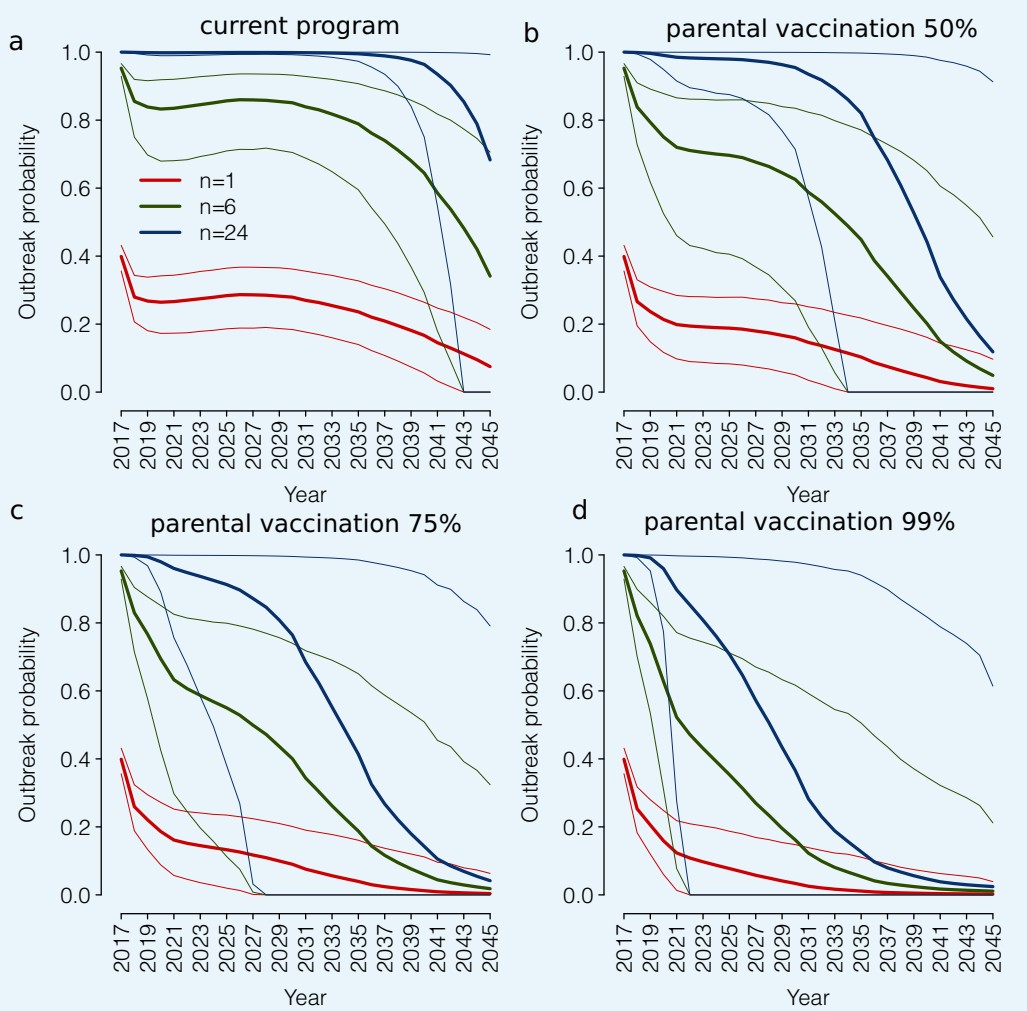

**Appendix 1—figure 5.** Outbreak probability under different assumptions relating to the number of imported cases over a year. (**a**) Outbreak probability (mean, bold lines, and 95% CI, thin lines) associated to n=1, n=6, n=24 yearly imported measles cases as estimated by the model for the period 2017–2045 under the current program. (**b**) As (**a**) but under the parental vaccination program with 50% coverage. (**c**) As (**a**) but under the parental vaccination program with 75% coverage. (**d**) As (**a**) but under the parental vaccination program with 99% coverage.

DOI: https://doi.org/10.7554/eLife.44942.028

According to our results, under the extreme assumption of only one measles importation over a year (*n* = 1), the outbreak probability in 2017 is around 40%. Under the current program, the outbreak probability would decrease over time (***Appendix 1—figure 5a***), however, it would be on average always greater than 0. Parental vaccination would instead result in an outbreak probability approximating 0 in 2035–2045, depending on the considered coverage level (***Appendix 1—figure 5c-d***). On the other hand, if we assume that the number of imported cases is around two per month (*n* = 24), the outbreak probability estimated in 2017 is ~1, and under the current program, it would remain around 1 up to 2035. The introduction of parental vaccination has instead the potential to decrease the outbreak probability to values lower than 20% by 2045 under all considered assumptions on the yearly number of imported cases.

## 2.Sensitivity analysis

### 2.1 Underreporting of the 2017 outbreak

In this sensitivity analysis, we assess the robustness of our estimates of the effective reproduction number when considering that undetected measles cases may have occurred in the 2017 outbreak. As our estimates of the effective reproduction number depend only on the growth rate and on measles natural history, we need to check the robustness of the estimated exponential growth rate $r$ of the 2017 outbreak to the case reporting rate. The current degree of reporting of measles in statutory notifications in Italy is unknown. However, a previous study highlighted that in the year 2000, the number of cases reported to a network of voluntary primary care pediatricians was 3.9 times higher than that reported in statutory notifications (*Ciofi Degli Atti et al., 2002*). This would suggest that only 25% of actual cases were reported in statutory notifications. The reporting rate of measles is likely to have improved in the past decades, especially in epidemic situations. However, in the absence of updated estimates, we decided to assess the robustness of the growth rate $r$ of the observed epidemic when considering a worst case scenario in which only 25% of cases were reported in the 2017 Italian outbreak.

To do this, we generate 1000 synthetic time-series of measles cases that may have occurred in 2017, by using an MCMC approach applied to the Poisson likelihood of observing the actual number of measles cases reported in each week of the year. Specifically, we assumed that for each week $w$, the likelihood of observing $c$ measles cases reported to the Italian National Institute of Health is:

$$L_w(c|\alpha) = \frac{e^{-\rho\alpha}(\rho\alpha)^c}{c!}$$

where $\rho$=0.25 is the reporting rate, $c$ is the number of measles cases reported to the Italian National Institute of Health in week $w$ of the year 2017 (*Italian National Institute of Health, 2017*), and $\alpha$ is the overall number of measles cases in Italy in week $w$ of the year 2017. The weekly number of measles cases obtained through this procedure (mean and 95% CI) is shown in *Appendix 1—figure 6*. Estimates of the exponential growth rate $r$ are derived by fitting a linear model to the logarithm of the cases obtained in each synthetic time series (illustrative examples of the obtained time series are shown in *Appendix 1—figure 7*). The resulting estimate of the exponential growth rate associated with the 2017 outbreak is 0.29 (95% CI: 0.22–0.37), similar to that obtained when only reported cases are used: 0.29 (95% CI: 0.25–0.33). We thus conclude that the estimates of the growth rate are sufficiently robust with respect to the possible underreporting of cases during the 2017 outbreak.

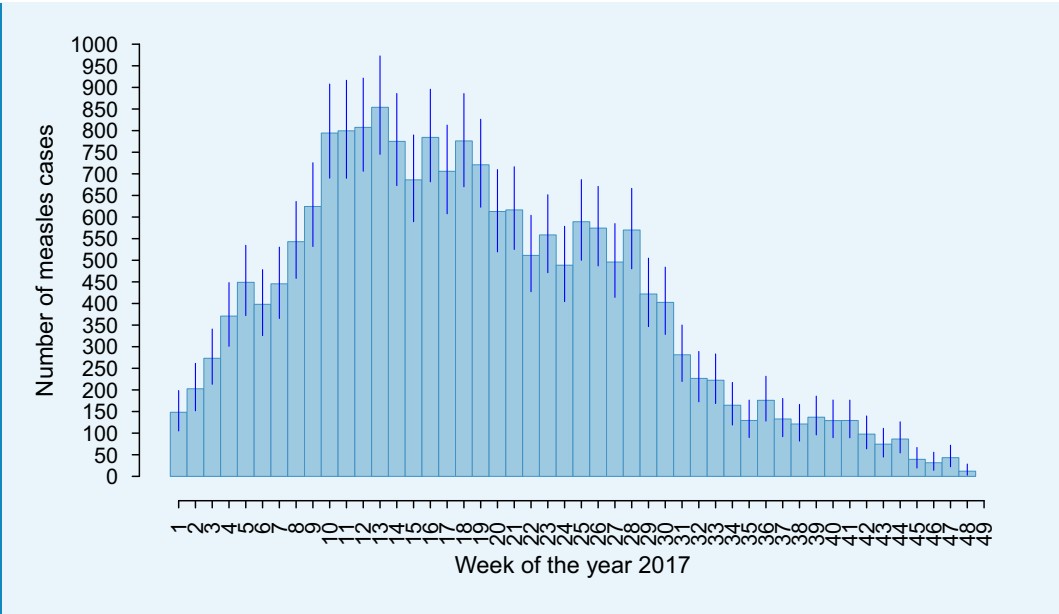

**Appendix 1—figure 6.** Weekly number of overall measles cases (mean and 95% CI) per week of the year 2017 as obtained when assuming a reporting rate of 25%.

DOI: https://doi.org/10.7554/eLife.44942.029

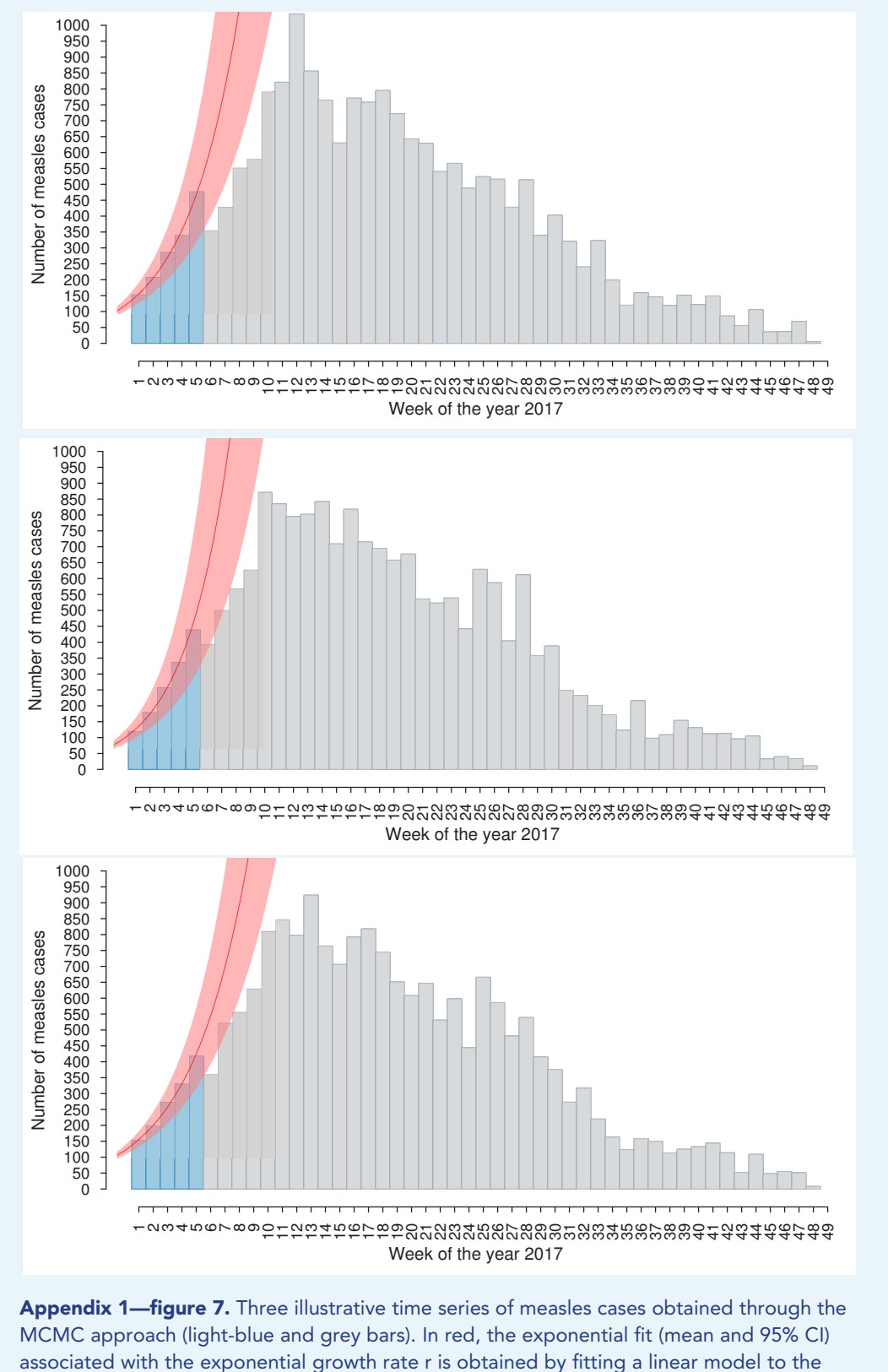

**Appendix 1—figure 7.** Three illustrative time series of measles cases obtained through the MCMC approach (light-blue and grey bars). In red, the exponential fit (mean and 95% CI) associated with the exponential growth rate r is obtained by fitting a linear model to the logarithm of the estimated weekly cases in the early epidemic phase (defined by light-blue bars).

DOI: https://doi.org/10.7554/eLife.44942.030

## 2.2 Measles generation time

In the main text, we assume an average latent period $1/\omega = 6.5$ days and an average infectious period of $1/\gamma = 7.5$ days, thus resulting in an average generation time of 14 days. In this sensitivity analysis, we assess the robustness of the estimates of the effective reproduction number over the 2017–2045 period when assuming a shorter or longer generation time for measles—namely, 10 and 18 days, respectively. For the sake of simplicity, we assume that the relative duration of the latent and infectious period is the same as in the main text.

According to the obtained results, when a generation time of 10 days is considered, the effective reproduction number estimated in 2017 is 1.46 (95% CI: 1.38–1.52) and it would decrease down to 0.96 (95% CI: 0.85–1.08) in 2045 under the current program (*Appendix 1—figure 8a-b*). Under this program, measles elimination would be achieved on average in 2043. The obtained results suggest that the introduction of parental vaccination (coverage of 50%, 75% and 99%) on top of the current program would anticipate measles elimination in 2034, 2027 and 2022, respectively.

Conversely, as shown in *Appendix 1—figure 8c-d*, when assuming a generation time of 18 days, the current program is insufficient to achieve measles elimination by 2045 (0.1% of model realizations). The introduction of parental vaccination at 99% of coverage on top of the current program (representing a best case scenario) would decrease the effective reproduction number to 0.97 (95% CI:0.81–1.18) in 2045 and measles elimination by this year is achieved in 71.4% of model realizations.

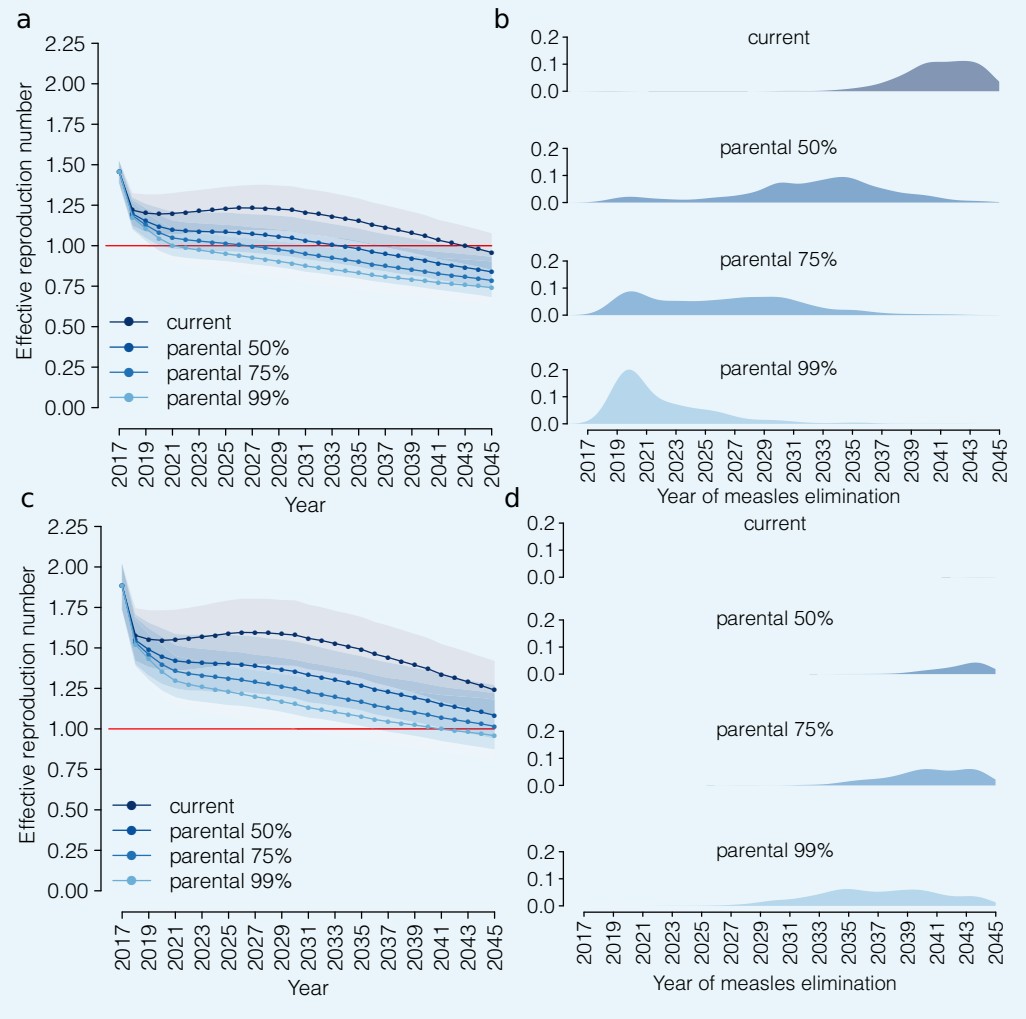

**Appendix 1—figure 8.** Future trends in measles elimination under different assumptions on the

generation time. (**a**) Mean effective reproduction number over time, as estimated by the model under the 'current' vaccination program and under different coverage scenarios for the 'parental vaccination' program when assuming a generation time lasting 10 days on average. Shaded areas represent 95% CI associated with model estimates. The red line represents the measles elimination threshold; elimination is achieved when the effective reproductive number is smaller than 1. (**b**) Probability associated with the different times of measles elimination, as obtained from 1000 model realizations under the 'current' vaccination program and under different coverage scenarios for the 'parental vaccination' when assuming a generation time lasting 10 days on average. (**c**) As (**a**) but when assuming a generation time lasting 18 days on average. (**d**) As (**b**) but when assuming a generation time lasting 18 days on average.
DOI: https://doi.org/10.7554/eLife.44942.031

## 2.3 Mixing patterns of the population

Estimates of the effective reproduction number presented in the main text are based on heterogeneous mixing patterns by age estimated by *Mossong et al. (2008)*. In this sensitivity analysis, we assess the robustness of our results when:

 i. considering an alternative contact matrix estimated for Italy available in the literature (*Fumanelli et al., 2012*);

 ii. assuming that the population mixes fully at random (i.e., homogenous mixing assumption).

In the first case, for each year $y$ and each vaccination scenario, the effective reproduction number is computed in a manner that is analogous to the baseline analysis (Section 1.3), except for that we use the contact matrix estimated for Italy by *Fumanelli et al. (2012)*, which was obtained through the construction of a virtual population parameterized with detailed socio-demographic data.

Under the homogeneous mixing assumption, for each year $y$ and each vaccination scenario, the effective reproduction number can simply be computed as:

$$R_e(y) = q\, S(y)$$

where $q$ is a scale factor representing the basic reproductive number associated with measles, and $S(y)$ is the total fraction of susceptible individuals estimated by the model in year $y$ under the considered vaccination scenario. In both cases, the value of $q$ is derived as described in Section 1.3.

Estimates of the effective reproduction number over time obtained when assuming the alternative contact matrix estimated for Italy are qualitatively robust with respect to the predictions obtained under our baseline assumption. However, in this scenario, we observe a delay of measles elimination under all considered vaccination scenarios with respect to the results of the baseline analysis presented in the main text (*Appendix 1—figure 9*).

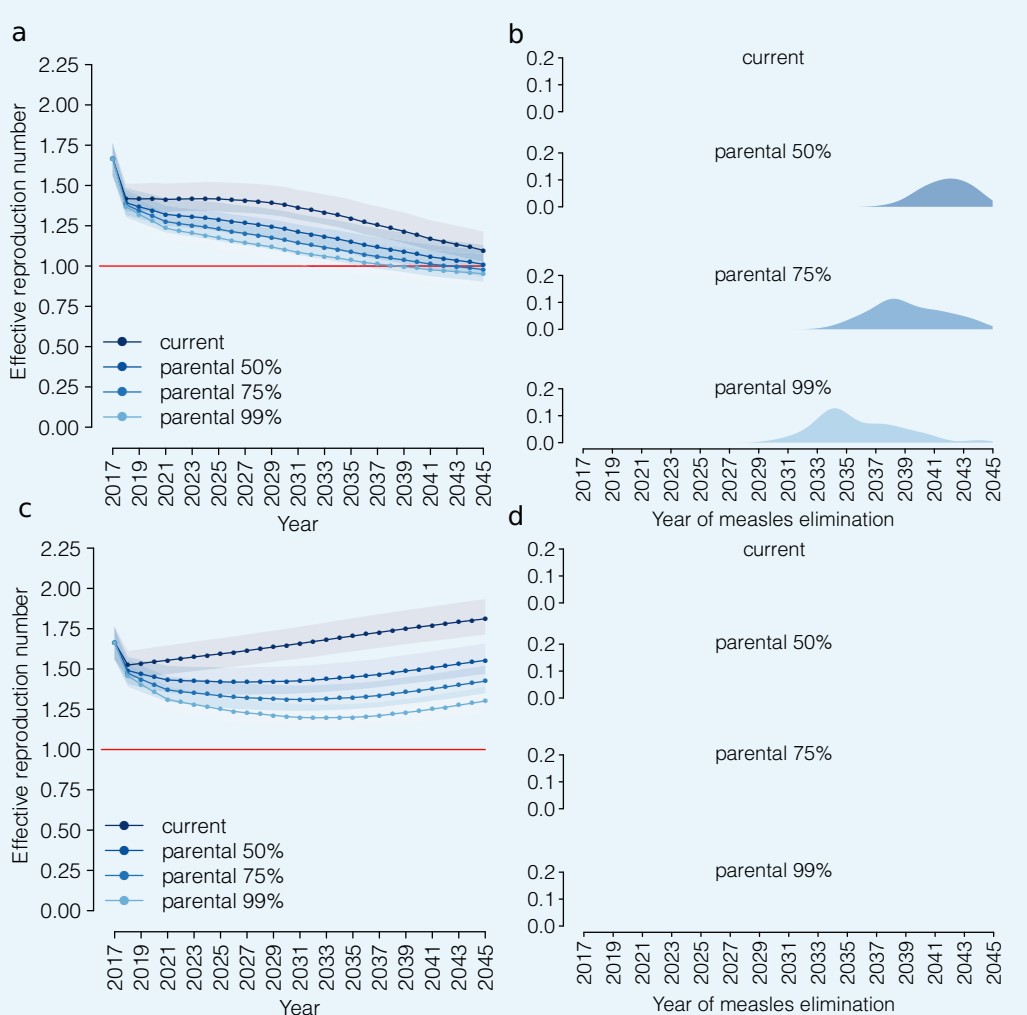

**Appendix 1—figure 9.** Future trends in measles elimination: population mixing. (**a**) Mean effective reproduction number over time, as estimated by the model under the 'current' vaccination program and under different coverage scenarios for the 'parental vaccination' program when assuming contact matrices estimated by *Fumanelli et al. (2012)*. Shaded areas represent 95% CI associated with model estimates. The red line represents the measles elimination threshold; elimination is achieved when the effective reproductive number is smaller than 1. (**b**) Probability associated with different time at measles elimination, as obtained from 1000 model realizations under the 'current' vaccination program and under different coverage scenarios for the 'parental vaccination' when assuming contact matrices as estimated by *Fumanelli et al. (2012)*. (**c**) As (**a**) but when assuming homogeneous mixing in the population. (**d**) As (**b**) but when assuming homogeneous mixing in the population.

DOI: https://doi.org/10.7554/eLife.44942.032

On the other hand, under the (hardly realistic) scenario of a population that mixes fully at random (i.e., the homogeneous mixing assumption), neither the current policy nor parental vaccination appear to be sufficient to achieve measles elimination by 2045. In particular, in this case, under the current program, after an initial drop in the effective reproduction number resulting from the catch-up campaign implemented in 2017, the effective reproduction number would steadily increase over time reaching 1.81 (95% CI: 1.71–1.93) in 2045 (see *Appendix 1—figure 9c*). The obtained results show that the introduction of parental vaccination can mitigate such an increase. However, even when assuming a 99% coverage for this program, the effective reproduction number is estimated to remain above the epidemic

threshold of 1 by 2045 and measles elimination is achieved in 0% of model realizations (*Appendix 1—figure 9d*).

## 2.4 Prodromal and exanthema phases

Estimates of the effective reproduction number discussed in the main text are based on the assumption that measles transmission follows an SLIR model, with a single infectivity phase lasting on average $1/\gamma = 7.5$ days (see Equation 1). We performed a sensitivity analysis to assess the robustness of our results when explicitly accounting for two distinct phases of infectivity, instead of a single one. Specifically, we assumed that, after an average latent period $1/\omega$, individuals enter first the prodromal phase (lasting on average $1/\gamma_1$) and then the exanthema phase (lasting on average $1/\gamma_2$). During both the prodromal and exanthema phases, individuals may transmit the virus at different rates. Under these assumptions, the effective reproduction number can be computed from the exponential growth rate $r$ as follows:

$$R_e^{2017} = \frac{1}{\left[(1-k)\left(\frac{\omega}{\omega+r}\right)\left(\frac{\gamma_1}{\gamma_1+r}\right) + k\left(\frac{\omega}{\omega+r}\right)\left(\frac{\gamma_1}{\gamma_1+r}\right)\left(\frac{\gamma_2}{\gamma_2+r}\right)\right]}$$

where, $k$ and $1-k$ represent the fraction of cases generated during the exanthema phase and prodromal phase, respectively. We assume that the average latency period is $1/\omega$=6.5 days, as in the main analysis, whereas the duration of infectivity is equally divided between the prodromal and exanthema phases, that is $1/\gamma_1$= $1/\gamma_2$ =3.75 days. We explore different scenarios for the relative contribution of each of the two infectivity phases to measles transmission (k in the range 10% to 90%). Obtained estimates of the effective reproduction number in 2017 vary between 1.49 (95% CI: 1.41–1.55), when we assume that 90% of secondary cases of a primary infector are generated during the prodromal phase ($k$=10%), and 1.67 (95% CI: 1.55–1.77), when 90% of cases are generated during the exanthema phase ($k$=90%).

From 2018 onwards, the yearly effective reproduction number is computed using the procedure described in Section 1.3. The estimates obtained for different values of $k$ are shown in *Appendix 1—figure 10* together with the probability associated with different times at measles elimination (left and right columns, respectively). Under all considered vaccination scenarios, the temporal patterns in the evolution of $R_e$ over the period 2018–2045 are generally robust.

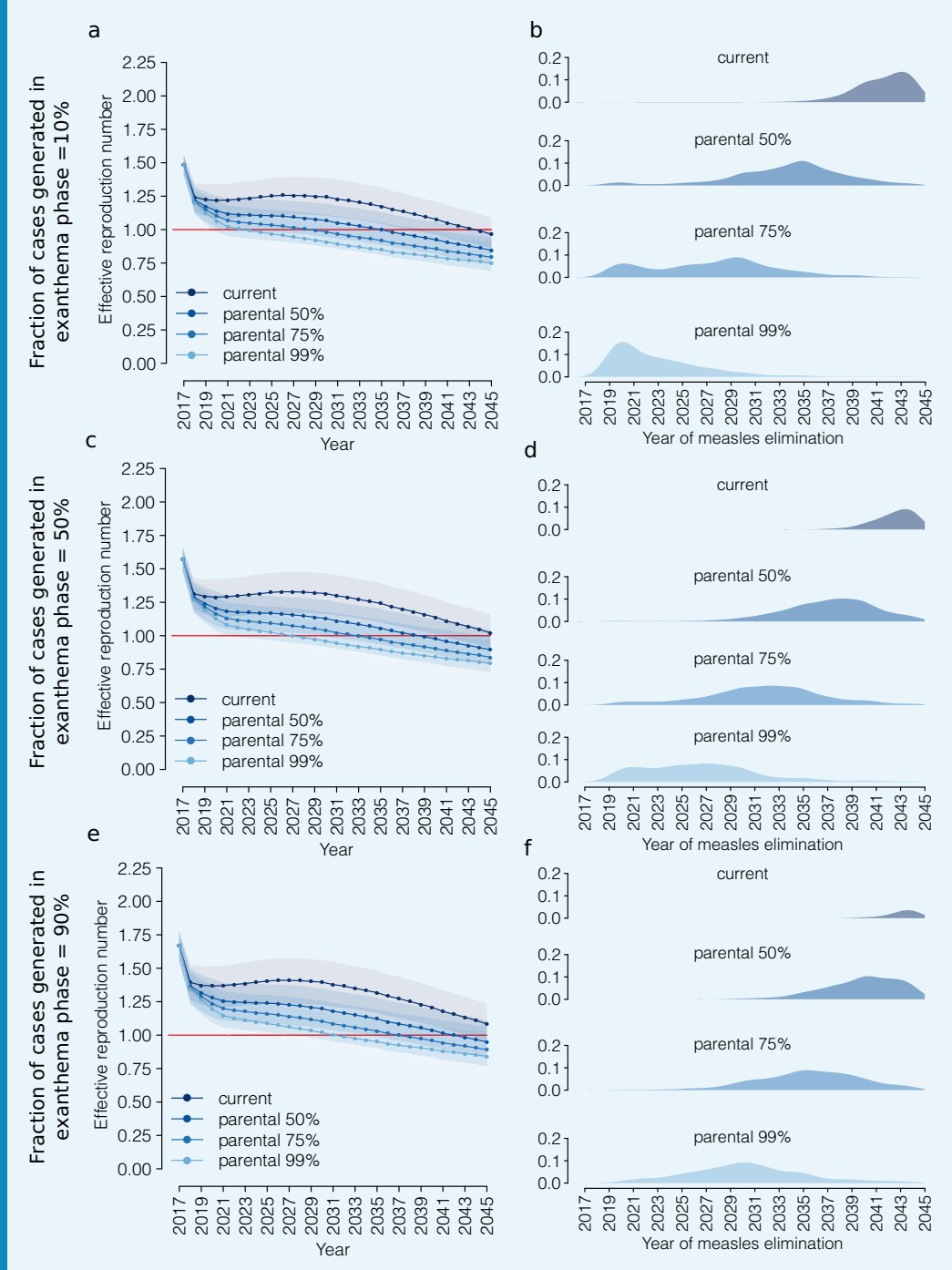

**Appendix 1— figure 10.** Future trends in measles elimination: prodromal and exanthema phases. Left column: mean effective reproduction number over time, as estimated by the model under the 'current' vaccination program and under different coverage scenarios for the 'parental vaccination' program. Shaded areas represent the 95% CI associated with the model estimates. The red line represents the measles elimination threshold; elimination is achieved when the effective reproductive number is smaller than 1. Right column: probability associated with different time at measles elimination, as obtained from 1000 model realizations under the 'current' vaccination program and under different coverage scenarios for 'parental vaccination'. For each vaccination scenario, the effective reproduction number in the year 2017 was computed by accounting for two distinct infectivity phases, prodromal and exanthema,

and by assuming that the latter contributes to a fraction k of secondary cases. Different rows correspond to different scenarios for the value of k: 10%, 50%, or 90%.
DOI: https://doi.org/10.7554/eLife.44942.033

The largest differences with respect to our baseline assumption are observed when most of the secondary cases are generated in the prodromal phase ($k = 10\%$). In this case, under the current program, measles elimination is estimated to occur before 2045 in 73.9% of model realizations compared to the 12.0% of model realizations with this result in the baseline analysis (*Appendix 1—figure 10 a-b*). Similarly, under parental vaccination at 50%, 75% and 99% of coverage, when most of secondary cases are generated in the prodromal phase, measles elimination is predicted to occur on average in 2035, 2029 and 2023, respectively. Conversely, the results obtained when assuming that most of the secondary cases are generated in the exanthema phase (k = 90%) are quantitatively comparable to those obtained in the baseline analysis (*Appendix 1—figure 10 e-f*).

## 2.5 Vaccination coverage at school entry

A large uncertainty surrounds the current estimates of coverage levels for measles vaccination at school entry. Therefore, we assessed the robustness of our estimates of the effective reproduction number over the 2017–2045 period when assuming a higher coverage for vaccination at school entry: 75% and 99%, instead of the 50% assumed in the analysis presented in the main text. In addition, we also considered a vaccination scenario for the current program in which the coverage at school entry is set to 0%. The latter scenario corresponds to the vaccination program that was in place prior the introduction of the new regulation in 2017 (denoted here as the 'past program'), which consisted of the recommendation of routine vaccination in a two-dose schedule. For all scenarios considered, coverage levels for the first and second dose routine vaccinations are fixed at 85% and 83%, respectively.

The obtained results suggest that, in the absence of the new regulation, the effective reproduction number would increase to 2.01 (95% CI: 1.83–2.28) in 2045 (past program – *Appendix 1—figure 11a-b*) and measles elimination would not be achieved by 2045. On the other hand, an improvement of measles vaccination coverage at school entry to 75% or 99% would result in a fall in the average effective reproduction number to levels below the epidemic threshold of 1 in 2042 and 2039, respectively (green and purple in *Appendix 1— figure 11a*). In addition, the percentage of simulations resulting in measles elimination before 2045 would also increase with respect to the baseline analysis: from 12% to 89.2% and from 12% to 98.8% when school entry vaccination is considered at 75% and 99% of coverage, respectively (*Appendix 1—figure 11b*). Furthermore, the obtained results confirm that the introduction of parental vaccination on top of the current program would accelerate progress towards measles elimination. In particular, according to our simulations, the introduction of parental vaccination (coverage of 50%, 75% and 99%) on top of the current program with a vaccination coverage at school entry of 75% would result in measles elimination in 2036, 2031 and 2026, respectively (*Appendix 1—figure 11 c-d*), compared to 2042, 2037 and 2031 obtained in the main analysis. If vaccine uptake at school entry were at 99% of coverage, parental vaccination (coverage of 50%, 75% and 99%) would result in measles elimination in 2033, 2028 and 2023, respectively (*Appendix 1—figure 11 e-f*).

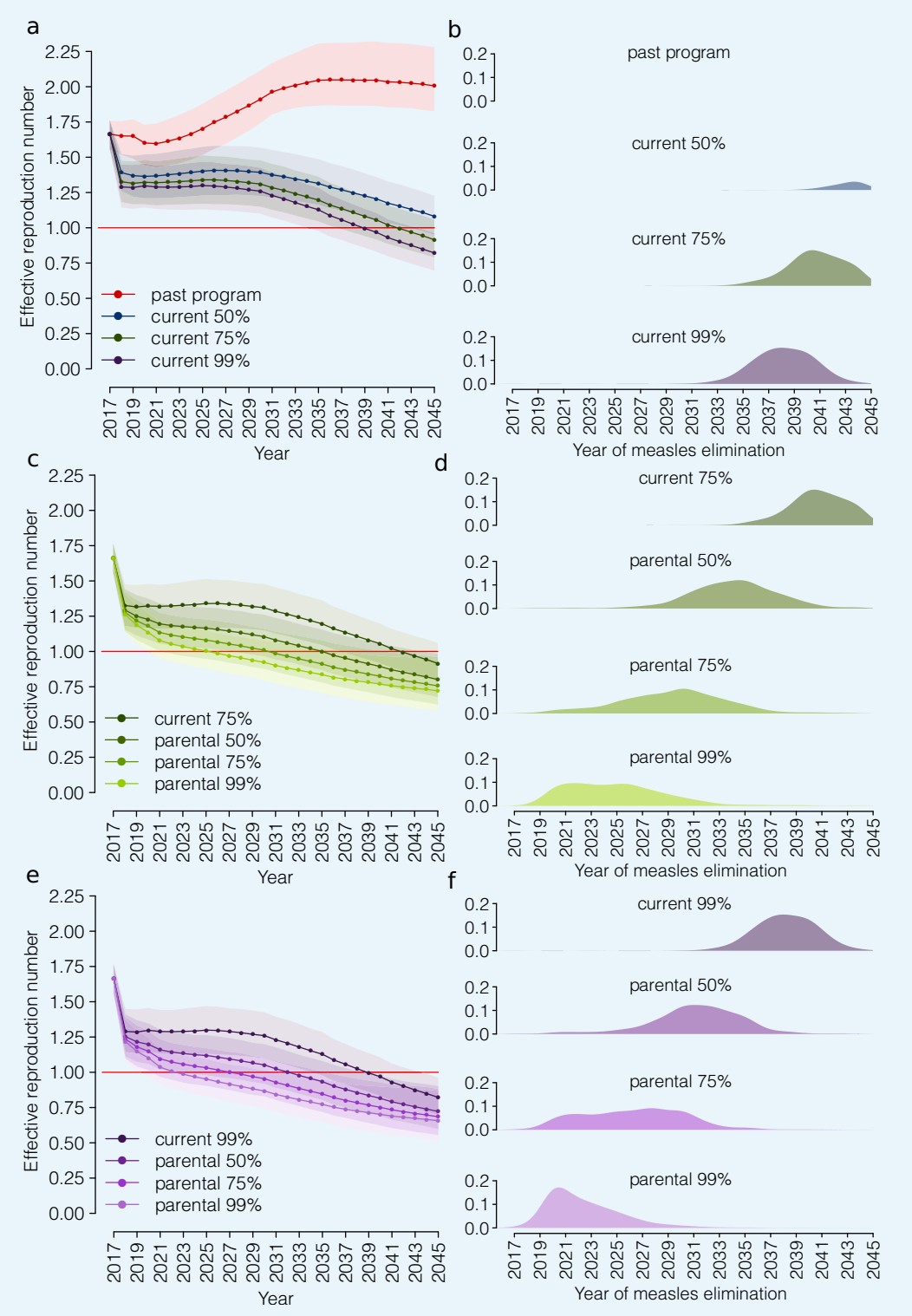

**Appendix 1—figure 11.** Future trends in measles elimination: different levels of vaccination coverage for the 'current' program. (**a**) Mean effective reproduction number over time, as estimated by the model under different coverage scenarios for the 'current' vaccination program: 0% (vaccination program in place before 2017), 50% (baseline), 75% and 99%. Shaded areas represent the 95% CI associated with the model estimates. The red line represents the measles elimination threshold; elimination is achieved when the effective reproductive number is smaller than 1. (**b**) Probability associated with different times at measles elimination, as obtained from 1000 model realizations under different coverage

scenarios for the 'current' vaccination program: 0%, 50% (baseline), 75% and 99%. (c) As (a) but as obtained under the 'current' vaccination program (coverage 75%) and under different coverage scenarios for 'parental vaccination'. (d) As (b) but as obtained under the 'current' vaccination program (coverage 75%) and under different coverage scenarios for 'parental vaccination'. (e) As (a) but as obtained under the 'current' vaccination program (coverage 99%) and under different coverage scenarios for 'parental vaccination'. (f) As (b) but for as obtained under the 'current' vaccination program (coverage 99%) and under different coverage scenarios for 'parental vaccination'.

DOI: https://doi.org/10.7554/eLife.44942.034

