## [Decision Letter]

Thank you for submitting your article "Parental and mandatory school entry vaccination to reduce measles immunity gaps in Italy" for consideration by *eLife*. Your article has been reviewed by Neil Ferguson as the Senior Editor, a Reviewing Editor, and three reviewers. The following individuals involved in review of your submission have agreed to reveal their identity: Felicity Cutts (Reviewer #1).

The reviewers have discussed the reviews with one another and the Reviewing Editor has drafted this decision to help you prepare a revised submission.

Summary:

This paper presents a modelling study of the impact of the new mandatory vaccination policy implemented in Italy since 2017 at school entry. The authors also explore the potential supplementation of the program with a campaign targeting parents. The impact is assessed using an individual-level model of household composition and simulating different vaccination policies up to 2045. The age specific susceptibility levels over the years are used to derived associated effective reproduction numbers and probabilities of outbreak.

Essential revisions:

The reviewers raise a number of concerns that must be adequately addressed before the paper can be accepted. Some of the required revisions will likely require further simulations and restructuring manuscript.

1) Restructure the manuscript in order to focus more on the third strategy (R+S+P).

2) Discuss what could be different strategies to reach older age groups (i.e. cohort with low level of immunity due to natural infection or vaccination) and their likely impact.

3) Ensure that the coverages described in the methods and used for the simulations are clearly defined in particular specify the denominators.

4) Do a sensitivity analysis regarding under reporting and size of 2017 outbreak; discuss plausibility of estimate in the Discussion section.

5) Specify if the age profile of observed cases used for validation of the model is based on laboratory cases.

6) Provide the ISTAT data as supplementary material to ensure permanency and provide a description of the assumptions of the model used for forecasting the demographic profile until 2045, this could be added to the Appendix.

7) Please incorporate uncertainty regarding demographic projections, estimation of r, contact matrices and initial susceptibility.

8) Include in the results a section on the sensitivity analysis based on results presented in the Appendix.

9) Carry additional sensitivity analysis with regards to estimation of R0.

10) Discuss the impact spatial heterogeneity in term of the predictions.

11) Please discuss the assumption behind the equation of the probability of an outbreak as a function of R_e_. A description in the Appendix of how the result is derived would be good. Is it the probability for a single introduction? How does it translate in terms of probability of a outbreak over a season? What is the impact of multiple reintroduction of measles?

---

## [Author Response]

Summary:This paper presents a modelling study of the impact of the new mandatory vaccination policy implemented in Italy since 2017 at school entry. The authors also explore the potential supplementation of the program with a campaign targeting parents. The impact is assessed using an individual-level model of household composition and simulating different vaccination policies up to 2045. The age specific susceptibility levels over the years are used to derived associated effective reproduction numbers and probabilities of outbreak.Essential revisions:The reviewers raise a number of concerns that must be adequately addressed before the paper can be accepted. Some of the required revisions will likely require further simulations and restructuring manuscript.

We thank the reviewers and the editors for taking the time to review our manuscript and for the constructive feedback provided. We enclose here a point-by-point response to each of the reviewers’ comments. We have done our best to address all comments and suggestions by the reviewers. In particular, the major changes can be summarized as follows:

- We have restructured the manuscript to better highlight the impact of introducing parental vaccination on top of the current strategy. In particular, we decided to focus the main text on strategy “R+S+P” by assuming a baseline vaccination coverage at school entry of 50% on the basis of the latest available estimates on the impact of the new regulation in Italy. Alternative scenarios with different coverage for vaccination at school entry (namely, 0%, 75%, and 99%) were moved to the Appendix and are now presented as a sensitivity analysis. The title has been amended to reflect this change. The figures of the manuscript have been updated as well.

- Model estimates now include the uncertainty regarding: (i) the demographic projections of the age structure of the Italian population over 2018-2045; (ii) the age-specific measles immunity profile; (iii) the growth rate r of the 2017 measles epidemic; iv) the Italian contact matrices.

-We have added: (i) a sensitivity analysis to assess the robustness of the estimates with respect to possible underreporting and uncertainty about the size of the 2017 Italian outbreak; and (ii) a sensitivity analysis on the effective reproduction number. A new subsections entitled “Sensitivity analysis” has also been added to the main text.

- We have specified in the text all missing details about the data and model assumptions.

-The Italian population forecasts over 2018–2045 used in our analysis are now provided as a Supplementary file, as long as a brief description of methods adopted by ISTAT to produce these forecasts.

- We moved to the Appendix results concerning the probability of experiencing a measles outbreak and added an analysis showing how it depends on the number of measles introductions over the course of a season.

1) Restructure the manuscript in order to focus more on the third strategy (R+S+P).

In light of the reviewers’ comment, we have restructured the manuscript in order to highlight the impact of introducing parental vaccination on top of the current regulation (R+S). In particular, according to the latest available estimates, measles vaccination coverage in the cohort 2014 has increased from 87.3% in 2016 to 94.4% as of June 2018, suggesting that the new regulation contributed to vaccinate about 56% of unvaccinated children in this cohort. In light of this, we decided to present in the main text results obtained under program R+S, including routine vaccination in a two-dose schedule and assuming a baseline vaccination coverage at school entry of 50%. We now refer to this program as “current program”. Regarding parental vaccination, we explore three different scenarios obtained by adding parental vaccination on top of the “current program” and assuming three different coverage levels (50%, 75%, and 99%), as in the original manuscript.

Alternative scenarios of the parental vaccination on top of the “current program” with different coverage for vaccination at school entry (75% and 99%) and the scenario including routine vaccination only (R) were moved to the Appendix and are now presented as sensitivity analyses. The figures of the manuscript have been updated accordingly.

We would like to thank the reviewers for providing this suggestion, as we do believe that the manuscript has now a much better flow and it clarifies the most important aspects of our analysis.

2) Discuss what could be different strategies to reach older age groups (i.e. cohort with low level of immunity due to natural infection or vaccination) and their likely impact.

This is an interesting point of discussion. In the new version of the manuscript, we discuss possible strategies alternative to parental vaccination that could be used to reduce measles immunity gaps in older age groups. An alternative possible intervention consists in extending mandatory vaccination to universities. Indeed, two doses of the measles vaccination are already required in several US states for attending colleges and universities, and all students have to document immunity to measles before registering for classes. In Italy, to date, no vaccinations are required to enrol at universities. A second possibility is represented by targeting health care workers (HCW), for which measles vaccination is recommended, but not mandatory in most European countries including Italy [Galanakis et al., 2014]. We do believe that specific interventions targeting this subpopulation should be a priority, given the high number of measles cases reported among them in the 2017 Italian outbreak (315 out of 5,098 cases) and their potential to amplify measles outbreaks. Possible interventions in this direction include the performance of catch-up campaigns for HCW or the introduction of a proof of immunity as a requirement for admissibility to employment as HCW. The latter policy is already in place in some Italian regions (e.g., Puglia and Emilia Romagna) [Maltezou et al., 2019].

To address this point, we added the following sentences to the Discussion:

“Beyond parental vaccination, alternative immunization strategies aimed at reducing residual susceptibility in adults may be considered as well. These may include the extension of mandatory vaccination at university entry – an intervention already implemented in different US states. Other immunization efforts may include the introduction of a proof of immunity as a condition for enrolment of health care workers (HCWs), for which measles vaccination is only recommended in most of European countries [Galanakis et al., 2014, Maltezou et al., 2019]. The need for improving vaccination coverage among HCWs is due to their potential in amplifying measles outbreaks and their higher risk of exposure to the virus as happened in the 2017 Italian outbreak where 7% of cases were HCWs [Maltezou et al., 2019].”

3) Ensure that the coverages described in the methods and used for the simulations are clearly defined in particular specify the denominators.

We apologize for being unclear on this point, that is now specified in the section Material and Methods section as follows:

“Coverage levels for the first and second dose of routine vaccination are assumed to be constant over time and set equal to the most recent estimates of measles vaccination coverage at the national level – 85% and 83%, respectively. In our simulation, the first dose is administered to children who have never been vaccinated and the second dose is administered to those who have only received one dose.”

[…]

“Specifically, the coverage level at pre-primary school entry represents the percentage of vaccine uptake among 3 years old children who have never been vaccinated, whereas the coverage at primary school entry represents the percentage of vaccine uptake among children who have received less than two doses.”

[…]

“In our simulation, parental vaccination is offered only once to each household, the first time they bring one of their children to get vaccinated under the current policy. In particular, we evaluate the impact of parental vaccination under three different coverage scenarios: 50%, 75% and 99%. These percentages represent the proportion of parents who get vaccinated with this strategy among all eligible parents, whose exact amount depends on the coverage for childhood vaccination programs. We assume that one single vaccine dose is offered to each parent during parental vaccination.”

4) Do a sensitivity analysis regarding under reporting and size of 2017 outbreak; discuss plausibility of estimate in the Discussion section.

Following the reviewers’ suggestion, we have added a sensitivity analysis on underreporting and size of the 2017 outbreak. The current degree of reporting of measles in statutory notifications is unknown. A previous study estimating Italian measles incidence rates in the year 2000 from cases reported to a network of voluntary primary care paediatricians and from statutory notifications found that the former was 3.9 times higher than the latter [Ciofi degli Atti et al., 2002]. This would suggest a degree of reporting in statutory notifications of about 25%. The reporting rate of measles may have likely improved in the last decades. However, in the absence of updated estimates, we take this value as a worst-case scenario and assess the robustness of our results when accounting for a reporting rate of 25%.

In the new version of the manuscript, we have included an additional analysis where we estimate the attack rate and the growth rate associated with 1,000 synthetic time-series of measles infection cases generated on the basis of a MCMC approach applied to the likelihood of observing the original time series of reported measles cases when assuming a reporting rate of 25%. Under this assumption on measles reporting, the estimated overall number of measles cases in 2017 is 19,224 (95%CI: 18,724-19,745 vs 5,098 reported).

Remarkably, even when considering this worst case scenario on measles reporting, the estimated exponential growth rate obtained is 0.29 (95% CI: 0.21-0.37), similar to the one obtained when only reported cases are used: 0.29 (95%CI: 0.25-0.33). Since estimates of the effective reproduction number depend only on the growth rate and on measles natural history, this suggests that our results are robust with respect to reporting rate (and size) of the 2017 measles outbreak.

Methodological details on this new sensitivity analysis are reported in the revised version of the Appendix.

The outcome of this sensitivity analysis is commented in the Results section as follows:

“Finally, when considering an extreme scenario where only 25% of measles cases were reported during the 2017 outbreak, we estimate the exponential growth rate to be 0.29 (95% CI: 0.21–0.37), similar to the one obtained when only reported cases are used: 0.29 (95% CI: 0.25–0.33). Since estimates of the effective reproduction number depend only on the growth rate and on measles natural history, this suggests that our results are robust with respect to reporting rate (and size) of the 2017 measles outbreak.”

5) Specify if the age profile of observed cases used for validation of the model is based on laboratory cases.

We apologize for the lack of detail on this. The data used for model validation consist in the age-specific number of suspected measles cases reported to the National Health Institute.

In order to clarify this point, we have modified the caption of Figure 1 as follows:

“Age distribution of susceptible individuals at the beginning of 2017 as simulated in our model (orange) and age distribution of suspected measles cases as reported during the year 2017 to the National Measles and Rubella Integrated Surveillance System (green).”

Moreover, we have modified the following sentence in Introduction to update the estimated number of measles cases reported in 2017 and specify that 4,042 out of 5,098 suspected cases were lab-confirmed measles cases.

“In 2017, Italy experienced one of the largest measles outbreaks occurred during the last decade in the European Region with four deaths and 5,098 cases, 4,042 of which were confirmed by positive laboratory results.”

6) Provide the ISTAT data as supplementary material to ensure permanency and provide a description of the assumptions of the model used for forecasting the demographic profile until 2045, this could be added to the Appendix.

As suggested, the ISTAT data on the projections of the Italian population by age over the period 2017–2045 used in our simulations is now included Supplementary file 2.

Moreover, the procedure adopted by the ISTAT (Billari et al., 2012) to provide demographic forecasts of the population between 2018 and 2045 is now summarized in the Appendix as follows:

“Projections of the Italian population used in this study (see Supplementary file 2) are based on different stochastic realizations of official forecasts as provided by ISTAT and obtained through a method introduced in the literature by Billari and colleagues in 2012. This method relies on the framework of the so-called “random-scenario approach”, which is based on a series of subsequent expert-based conditional evaluations on the future evolution of different demographic indicators, given the values of the indicators at previous time points. Component-specific forecasts are combined and applied to an initial population (2017) to obtain different projections of the age-structure and overall size of the Italian population between 2018 and 2060.”

7) Please incorporate uncertainty regarding demographic projections, estimation of r, contact matrices and initial susceptibility.

We would like to thank the reviewers for pointing out this important methodological aspect. In the new version of the main text, we now present and discuss epidemiological results associated with different trajectories of the effective reproduction number (R_e_) as obtained by taking into account the uncertainty on:

1) initial measles susceptibility in 2017 as estimated by Trentini et al., 2017;

2) demographic trajectories of the Italian population age-structure as provided by available projections for the period 2017–2045;

3) contact patterns as resulting from a bootstrap procedure applied to the Italian POLYMOD contact matrix;

4) exponential growth rate r associated with the 2017 measles epidemic as resulting from the linear regression analysis.

Please note that in the original version of the manuscript we were using the contact matrix estimated for Italy by Fumanelli et al., 2012, which were obtained through the construction of a virtual population parameterized with detailed socio-demographic data. However, in order to appropriately include in our estimates, the uncertainty surrounding age-specific contact patterns observed in Italy, we have now considered the POLYMOD contact matrix for Italy [Mossong et al., 2008]. In particular, this allowed us to generate 1,000 bootstrapped contact matrices for Italy based on the publicly available individual contact diaries of the POLYMOD study.

Please note, results obtained in the original manuscript by using the contact matrix estimated by Fumanelli et al., have now been included as a sensitivity analysis on model results.

In order to better clarify all the different sources of uncertainty included in our estimates we have added the following sentence in the Materials and methods section:

“Results presented in this paper are based on 1,000 different model realizations for each vaccination scenario and include uncertainty regarding: the demographic projections of the age structure of the Italian population over 2018–2045; the age-specific measles immunity profiles estimated for Italy for 2017; the estimated growth rate r of the 2017 measles epidemic; and the age-specific mixing patterns of the Italian population.”

A description on how the different sources of uncertainty have been incorporated is also provided in the new version of the Appendix.

8) Include in the results a section on the sensitivity analysis based on results presented in the Appendix.

A new subsection “Sensitivity analysis” discussing the results of the performed sensitivity analyses has been added at the end of the Results section. In particular, we assess the robustness of our results with respect to:

1) Estimates of the effective reproduction number by considering scenarios with higher coverage for measles vaccination at school entry (75% and 99%, instead of 50% as assumed in our baseline analysis), either combined with parental vaccination or not.

2) Estimates of the effective reproduction number by assuming longer or shorter generation time (10 and 18 days, as compared with the 14 days assumed in our baseline analysis).

3) Estimates of the effective reproduction number under different assumptions on population mixing patterns in the form of an alternative contact matrix estimated for Italy [Fumanelli et al., 2012] and under the homogeneous mixing assumption.

4) Estimates of the effective reproduction number by explicitly accounting for two distinct phases of infectivity, instead of a single one.

5) Estimated exponential growth *r* when including possible under-reporting of cases for the 2017 outbreak (see reviewers’ comment #4).

Please note that sensitivity analysis 1, 3, and 4 were already present in the original version of the manuscript, but that new simulations have been performed to incorporate the different sources of uncertainty that are now accounted for in the baseline analysis (as detailed above – reviewers comment #7).

Details regarding the performed sensitivity analyses have been added in the Appendix but also mentioned in the main.

9) Carry additional sensitivity analysis with regards to estimation of R0.

In our modelling analysis, the estimate of the (effective) reproduction number depends on the estimated growth rate of the 2017 outbreak and on measles natural history. As such, we performed a new sensitivity analysis aimed at evaluating the estimates’ robustness to different assumptions on the duration of the generation time. Specifically, we considered a shorter and longer generation time (10 and 18 days) with respect to the value assumed in the baseline analysis (14 days).

We have now added a section at the end of the Results section in which we present estimates obtained through this sensitivity analysis:

“The assumption of a shorter or longer generation time would affect model estimates of the effective reproduction number over time. In particular, under parental vaccination at 50% of coverage, a shorter (longer) generation time would result in an anticipation (delay) of the timing at measles elimination, which is estimated to occur before 2045 in 99.2% (16.8%) of model realizations. When a generation time lasting 18 days is considered, the current policy, at current coverage levels, results instead insufficient to achieve measles elimination by 2045 in 99.9% of model realizations.”

Additional details and figures on the sensitivity analysis performed have also been added in the Appendix.

10) Discuss the impact spatial heterogeneity in term of the predictions.

We thank the reviewers for this useful comment, which allowed us to clarify a potential limitation of our analysis. Indeed, available epidemiological data and evidence show that Italian regions have experienced different immunization schedules and vaccination coverage in the last decades [Bonanni, 2015]. The new regulation on mandatory vaccination will hopefully promote a harmonization in the vaccine offer and uptake among children. However, different measles immunity levels have been detected across regions (e.g., Rota et al., 2008), so that regions that have been characterized by lower coverage in the past could experience a delay in measles elimination with respect to the results presented in this paper. On the other hand, we believe that all Italian regions (and possibly other countries as well) would benefit from the introduction of parental vaccination aimed at reducing measles immunity gaps in adults. The following sentence has therefore been added in the Discussion section:

“Although the new regulation is expected to harmonize vaccine offer and uptake in Italy, significant regional differences in both immunization schedule and coverage were reported in the past. Regions characterized by a lower than national average vaccine uptake in the past may therefore experience a delay in measles elimination with respect to results presented in this work.”

11) Please discuss the assumption behind the equation of the probability of an outbreak as a function of R_e_. A description in the Appendix ion of how the result is derived would be good. Is it the probability for a single introduction? How does it translate in terms of probability of an outbreak over a season? What is the impact of multiple reintroduction of measles?

We apologize for the lack of detail on this point. The equation used in the original version of the manuscript represents the outbreak probability after a single reintroduction and holds for stochastic SIR epidemics models (see for instance discrete time Markov Chains [Allen, 2008]). The equation can be generalized as p=1-(1/R_e_)^n^ to obtain the probability of outbreak after *n* importations – either as separated events or not.

However, the estimated probability of an outbreak over a season can dramatically change under different assumptions on the numberof cases imported in the population over the considered period (e.g. in one year), which is a quantity that is difficult to properly estimate and forecast. As such, we acknowledge that the outbreak probability after a single reintroduction could represents a misleading measure of measles transmission potential. Thus, we decided to move results on this from the main text to the Appendix, by adding an appropriate discussion on how this probability would change when a different number of imported cases over a season is considered.

Please note that, as suggested, we added in the Appendix a description of the assumptions behind the equation p=1-1/R_e_, providing appropriate references and discussion on this point.